# Activity-dependent mitochondrial ROS signaling regulates recruitment of glutamate receptors to synapses

**Rachel L Doser[1,2], Kaz M Knight[1,3], Ennis W Deihl[1], Frederic J Hoerndli[1]\***

[1]Department of Biomedical Science, Colorado State University, Fort Collins, United States; [2]Department of Health and Exercise Sciences, Colorado State University, Fort Collins, United States; [3]Cellular and Molecular Biology Graduate Program, Colorado State University, Fort Collins, United States

**Abstract** Our understanding of mitochondrial signaling in the nervous system has been limited by the technical challenge of analyzing mitochondrial function in vivo. In the transparent genetic model *Caenorhabditis elegans,* we were able to manipulate and measure mitochondrial reactive oxygen species (mitoROS) signaling of individual mitochondria as well as neuronal activity of single neurons in vivo. Using this approach, we provide evidence supporting a novel role for mitoROS signaling in dendrites of excitatory glutamatergic *C. elegans* interneurons. Specifically, we show that following neuronal activity, dendritic mitochondria take up calcium ($Ca^{2+}$) via the mitochondrial $Ca^{2+}$ uniporter (MCU-1) that results in an upregulation of mitoROS production. We also observed that mitochondria are positioned in close proximity to synaptic clusters of GLR-1, the *C. elegans* ortholog of the AMPA subtype of glutamate receptors that mediate neuronal excitation. We show that synaptic recruitment of GLR-1 is upregulated when MCU-1 function is pharmacologically or genetically impaired but is downregulated by mitoROS signaling. Thus, signaling from postsynaptic mitochondria may regulate excitatory synapse function to maintain neuronal homeostasis by preventing excitotoxicity and energy depletion.

**\*For correspondence:** frederic.hoerndli@colostate.edu

**Competing interest:** The authors declare that no competing interests exist.

## Editor's evaluation

This study examines an interplay between synaptic mitochondria and glutamate receptor exocytosis in *C. elegans*. Collectively, the solid results support the idea that mitochondrial function influences receptor dynamics at postsynaptic sites. This is important because tight control of synaptic function likely integrates several mitochondrial functions: energy production, calcium buffering, and (here) reactive oxygen species signaling.

## Introduction

As the predominant excitatory synapse type in the brain, glutamatergic synapses are important for organismal physiology and homeostasis as well as much of the brain's processing. Plasticity, or the change in efficacy, of these synapses underlies learning and memory formation. Although presynaptic changes contribute to synaptic transmission strength, the number of ionotropic glutamate receptors, especially the α-amino-3-hydroxy-5-methyl-4-isoxazole (AMPA) subtype (AMPARs), at the postsynaptic membrane is a strong correlate of synaptic strength. Changes in synaptic expression of AMPARs is a calcium ($Ca^{2+}$)-dependent, multi-step process involving long-distance transport of the receptors by molecular motors (*Kim and Lisman, 2001*; *Setou et al., 2002*; *Hoerndli et al., 2013*; *Esteves da Silva et al., 2015*; *Hangen et al., 2018*; *Hoerndli et al., 2022*), delivery of AMPAR-containing vesicles

to synaptic sites (*Yang et al., 2008*; *Hoerndli et al., 2015*), exocytosis and endocytosis of AMPARs to the membrane (*Ehlers, 2000*; *Yudowski et al., 2007*), as well as reorganization of postsynaptic proteins and cytoskeletal architecture (*Choquet and Triller, 2013*; *Nakahata and Yasuda, 2018*; *Gutiérrez et al., 2021*).

The mechanisms underlying postsynaptic plasticity are metabolically demanding processes requiring the upregulation of mitochondrial metabolism to meet energy demands (*Wacquier et al., 2019*; *Faria-Pereira and Morais, 2022*). There is growing evidence that mitochondria are also important for other cellular functions, including regulation of gene expression, $Ca^{2+}$ homeostasis, inflammatory signaling, and lipid biogenesis (*Chae et al., 2013*; *Hirabayashi et al., 2017*). Interestingly, the generation of reactive oxygen species (ROS), such as superoxide and hydrogen peroxide, by the mitochondrial respiratory chain and other matrix proteins (*Angelova and Abramov, 2018*) is gaining traction as an essential signaling mechanism with many identified downstream effectors in neurons (*Sies and Jones, 2020*; *Hidalgo and Arias-Cavieres, 2016*). It has become clear that ROS act as a physiological signal (*Sies and Jones, 2020*) that is necessary for neuronal development (*Oswald et al., 2018b*), excitatory and inhibitory neurotransmission (*Biswas et al., 2022*), as well as synaptic plasticity (*Massaad and Klann, 2011*; *Oswald et al., 2018a*).

For instance, evidence accumulated over the last 25 years has demonstrated that ROS signaling is required for normal synaptic expression of AMPARs. Early evidence came from results suggesting abnormal plasticity of glutamatergic synapses, learning and memory when ROS are elevated or diminished (*Massaad and Klann, 2011*; *Klann et al., 1998*; *Knapp and Klann, 2002*; *Huddleston et al., 2008*). Since these studies, we and others have shown that ROS signaling can regulate the number of synaptic AMPARs via ROS-dependent regulation of AMPAR phosphorylation (*Lee et al., 2012*) or the long-distance transport and delivery of AMPARs to synapses (*Doser et al., 2020*; *Doser and Hoerndli, 2022*). Despite our understanding of several downstream effectors of ROS signaling, it is unclear when or where ROS signaling originates in neurons in vivo. As previously mentioned, ROS is predominantly generated as a by-product of mitochondrial respiration but is also produced by NADPH oxidase and peroxisome enzymes (*Sies and Jones, 2020*). Despite mitochondria being the major source of ROS, it has not been assessed in vivo if or how mitochondrial ROS (mitoROS) production is regulated by neuronal activity. In addition, mitochondria are positioned at pre- and postsynaptic sites (*Freeman et al., 2017*) where they likely contribute to synaptic function. However, our understanding of the roles mitochondria play at synapses has been limited by our ability to study mitochondrial function in vivo under physiological conditions.

The transparent nematode *Caenorhabditis elegans* is a powerful genetic model that has been widely accepted for studying mitochondrial function, $Ca^{2+}$ handling, and ROS signaling in vivo, especially in the context of aging and neurodegeneration (*Back et al., 2012*; *Petriv and Rachubinski, 2004*; *Morsci et al., 2016*; *Xu and Chisholm, 2014*; *Alvarez et al., 2020*). Additionally, *C. elegans* has been used extensively in neuroscience research (*Sengupta and Samuel, 2009*) due to their relatively simple nervous system composed of neurons whose gene expression and synaptic connections are completely mapped (*Cook et al., 2019*; *Taylor et al., 2021*). Importantly, most of the key players at glutamatergic synapses are conserved, including subunits of AMPARs and other glutamate receptor subtypes (*Maricq et al., 1995*), and are regulated in a similar fashion to their vertebrate orthologs (*Hangen et al., 2018*; *Hoerndli et al., 2015*; *Rongo and Kaplan, 1999*; *Widagdo et al., 2017*). Using *C. elegans* to study the regulation of glutamatergic synapses, we have shown that $Ca^{2+}$ signaling regulates transport and delivery of GLR-1, the *C. elegans* ortholog of the AMPAR subunit GluA1, to synapses. Moreover, our previous work revealed that ROS signaling interacts with $Ca^{2+}$ signaling in the cell body and dendrites to control the amount of GLR-1 transport and regulate synaptic delivery of GLR-1 (*Doser et al., 2020*). Thus, an interplay between ROS and $Ca^{2+}$ signaling at postsynaptic sites appears to be important for AMPAR localization to synapses, but the role of postsynaptic mitochondria in this process has not been addressed.

Here, using in vivo imaging and optogenetic tools in *C. elegans,* we assessed the role of postsynaptic mitochondria as signaling hubs that integrate neuronal activity and regulate AMPAR localization to synapses. We found that in response to neuronal activation, mitochondria take up $Ca^{2+}$, resulting in an increase in their ROS production. Most dendritic mitochondria were located in close proximity to clusters of surface-localized GLR-1, which are representative of postsynaptic sites. To demonstrate the functional relevance of activity-dependent mitoROS signaling, we show that activity-dependent

mitoROS production, requiring the mitochondrial Ca$^{2+}$ uniporter MCU-1, regulates transport, delivery, and recruitment of GLR-1 to synapses. Since the number of glutamate receptors at a synapse controls the efficacy of excitatory transmission, activity-induced mitoROS production may constitute a critical inhibitory feedback mechanism that balances neuronal excitability with cellular energy capacity.

## Results

### Activity-dependent mitochondrial Ca$^{2+}$ uptake regulates synaptic recruitment of GLR-1

As in vertebrates, the majority of neuronal activation in *C. elegans* are due to glutamatergic transmission. Activation occurs when glutamate is released from a presynaptic neuron that binds to and opens the cation pore of postsynaptic glutamate receptors, including AMPARs. Influx of cations into the postsynaptic neuron initiates opening of voltage-gated Ca$^{2+}$ channels that causes a rapid increase in cytoplasmic Ca$^{2+}$. This Ca$^{2+}$ activates a multitude of signaling cascades before being rapidly taken up by the endoplasmic reticulum and mitochondria or extruded to extracellular space (*Brini et al., 2013*). Mitochondria in various neuronal subtypes have discrete Ca$^{2+}$ handling capabilities (*Márkus et al., 2016*), so we first characterized mitochondrial Ca$^{2+}$ uptake in vivo in the neurites of the AVA glutamatergic interneurons. To do this, we co-expressed the light-sensitive cation channel ChRimson (*Klapoetke et al., 2014*) with the mitochondrial calcium indicator mitoGCaMP (*Ashrafi et al., 2020*) targeted to the inner mitochondrial matrix (*Figure 1A*, *Figure 1—video 1*). This combination of tools allowed us to measure Ca$^{2+}$ uptake by individual mitochondria following repetitive optical activation. It is important to note that our photoactivation protocol involved optical stimulation using a 1 s light pulse every 30 s (33.3 mHz), a rate that is similar to the spontaneous activity of AVA neurons (*Doser et al., 2020*). This assay revealed that there is diversity in Ca$^{2+}$ handling among dendritic mitochondria. Some mitochondria take up the most Ca$^{2+}$ upon the first optical activation (Mito 1; *Figure 1B and C*, *Figure 1—video 1*), whereas others uptake more Ca$^{2+}$ following the second or third stimulation (Mito 2; *Figure 1B and C*, *Figure 1—video 1*).

Ca$^{2+}$ entry into the matrix is gated by the Ca$^{2+}$-sensitive mitochondrial uniporter MCU-1 (*Baughman et al., 2011*), which is encoded by the *mcu-1* gene in *C. elegans*. We characterized the effect of the *mcu-1(ju1154)* loss of function allele (*Álvarez-Illera et al., 2020*) (hereafter called *mcu-1(lf)*) on activity-dependent mitochondrial Ca$^{2+}$ uptake by imaging mitoGCaMP in *mcu-1(lf)* (*Figure 1D–F*). We found that the amplitude of evoked mitoGCaMP events in *mcu-1(lf)* was drastically decreased compared to controls (*Figure 1D–F*). Additionally, the total mitoGCaMP activity, a combined measure of the amplitude and duration of all Ca$^{2+}$ events, was also reduced in *mcu-1(lf)* (*Figure 1F*). Due to the possibility of functional compensation in *mcu-1(lf)*, we also tested how acute treatment with the ruthenium compound Ru360, an MCU-1 blocker (*Woods et al., 2019*), alters activity-dependent mitochondrial Ca$^{2+}$ uptake. Following a 10 min treatment with Ru360, we observed a decrease in the amplitude and total activity of evoked mitoGCaMP events that were similar to *mcu-1(lf)*. To test the specificity of Ru360 for inhibiting Ca$^{2+}$ uptake via MCU-1, we treated *mcu-1(lf)* with Ru360 but did not detect additional inhibition of mitochondrial Ca$^{2+}$ uptake (*Figure 1D–F*). This Ru360 treatment suppressed mitochondrial Ca$^{2+}$ uptake out to 60 min post treatment (*Figure 1G–I*). This experiment showed that loss or inhibition of MCU-1 almost completely prevents activity-dependent mitochondrial Ca$^{2+}$ uptake.

While imaging mitochondrial-localized fluorescent indicators in the AVA glutamatergic interneurons, we observed that around 61% of mitochondria are in close proximity (<1 μm) to clusters of surface-localized GLR-1 (quantification not shown), indicative of postsynaptic sites, that were visualized using GLR-1 tagged with pH-sensitive GFP (SuperEcliptic pHlourin, SEP) on the N-terminal (*Figure 2A*). The regulation of mitochondrial function and signaling by Ca$^{2+}$ appears to be integral to synaptic function and plasticity (*Ashrafi et al., 2020*; *Stoler et al., 2022*; *Billups and Forsythe, 2002*; *Sun et al., 2013*; *Marland et al., 2016*), which led us to test if postsynaptic mitochondrial Ca$^{2+}$ uptake is required for normal GLR-1 localization to synapses. First, we quantified SEP::GLR-1 fluorescence in AVA dendrites in vivo to assess if the number of GLR-1 at synapses is altered by loss or inhibition of MCU-1. Initial observations revealed a dramatic increase in the fluorescence of SEP::GLR-1 puncta (indicative of synaptic sites) in *mcu-1(lf)* mutants (*Figure 2—figure supplement 1A*), but not puncta density (data not shown), suggesting more GLR-1 at synaptic sites. Acute Ru360 treatment slightly, but not significantly, increased the fluorescence of SEP::GLR-1 puncta along the AVA neurite

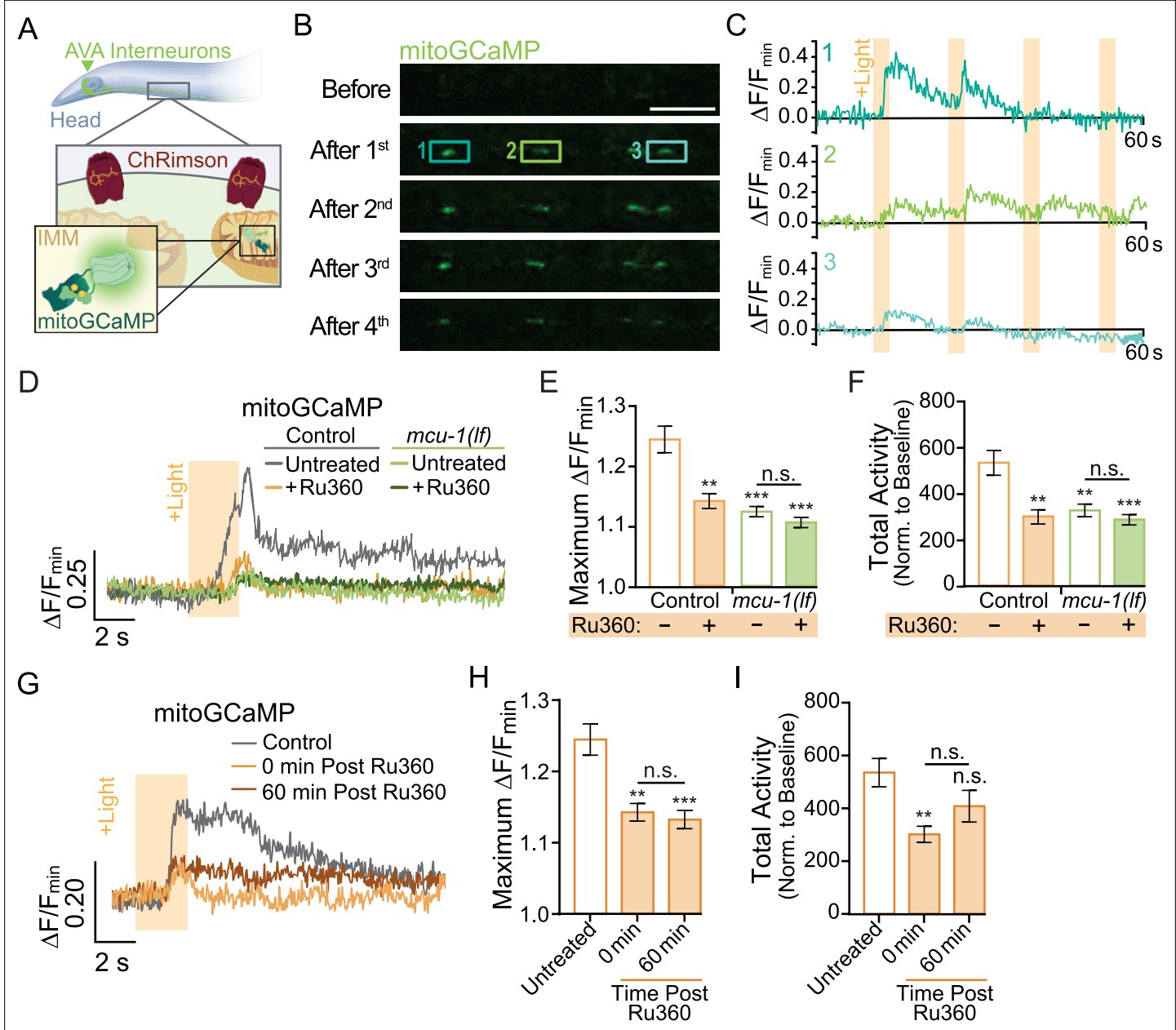

**Figure 1.** Neuronal activity causes mitochondrial Ca²⁺ uptake via MCU-1. (**A**) Illustration depicting transgenic expression and subcellular location of ChRimson and mitoGCaMP in the AVA neurons. (**B**) Representative images of mitoGCaMP fluorescence in a single Z-plane before and after four optical activations (strain: FJH 644). Scale bar = 5 µm. (**C**) Normalized mitoGCaMP fluorescence for the regions of interest in (**B**) during repetitive optical activation (+Light, 5 µW at 33.3 mHz). (**D**) Representative normalized mitoGCaMP traces (30 s) following optical stimulation (+Light) of the AVA neurons in worms pretreated with Ru360 and in untreated controls (strain: FJH 644) or *mcu-1(lf)* (strain: FJH 647). (**E**) Quantification of the maximum $\Delta F/F_{min}$ of mitoGCaMP events and (**F**) total mitoGCaMP activity during a 2.5 min recording of AVA neurons optically activated every 30 s (n ≥ 20 mitochondria from 5 to 8 animals per group). (**G**) Normalized mitoGCaMP fluorescence following optical stimulation (+Light) of the AVA neuron in untreated controls as well as Ru360-treated worms at 0 or 60 min post treatment. (**H**) Quantification of the average maximum $\Delta F/F_{min}$ of mitoGCaMP and (**I**) normalized total mitoGCaMP activity during a 2.5 min recording of AVA neurons optically activated every 30 s (n ≥ 20 mitochondria from 4 to 5 animals per group). Data is represented as mean ± s.e.m.; n.s., not significant, **p<0.005, ***p<0.0005 compared to controls using a one-way ANOVA with a Dunnett's test. Source data is available at https://doi.org/10.5061/dryad.0gb5mkm71.

The online version of this article includes the following video for figure 1:

**Figure 1—video 1.** MitoGCaMP fluorescence in the AVA neurites.

https://elifesciences.org/articles/92376/figures#fig1video1

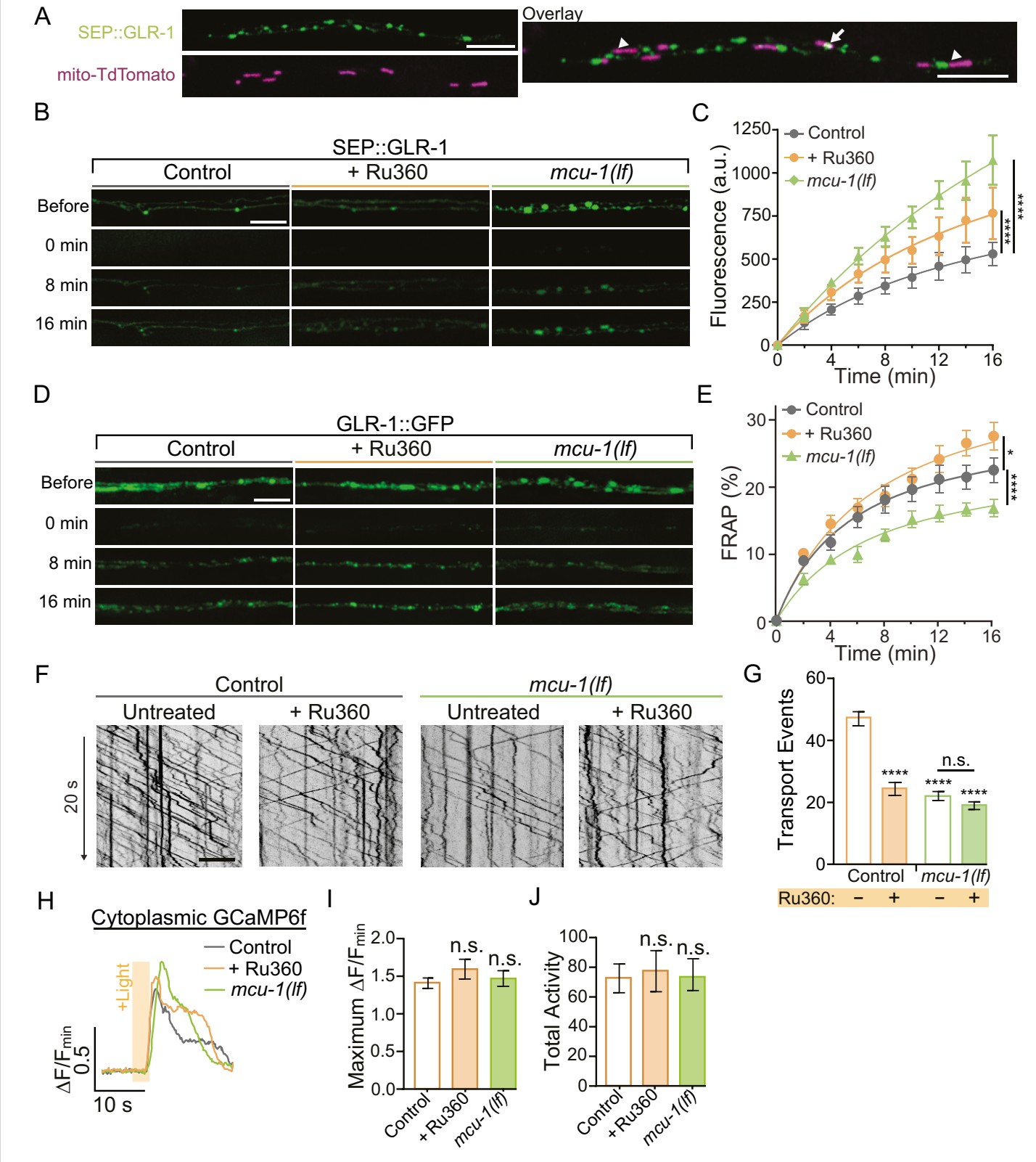

**Figure 2.** Decreased mitochondrial Ca²⁺ uptake affects transport and recruitment of GLR-1 to synapses. (**A**) Single Z-plane fluorescent images of mitochondria (mito-TdTomato) and surface-localized GLR-1 (SEP::GLR-1) showing mitochondria localized at (arrows) or adjacent to (arrowheads) SEP::GLR-1 puncta. (**B, D**) Representative images of (**B**) SEP::GLR-1 (strains: FJH 214 and FJH 638) or (**D**) GLR-1::GFP (strains: FJH 18 and FJH 576) fluorescence before, immediately after, 8, and 16 min post photobleach (PB). (**C**) Fluorescence (arbitrary units = a.u.) of SEP over 16 min post PB (n = 8

*Figure 2 continued on next page*

*Figure 2 continued*

animals per group). (**E**) Percent GFP fluorescence recovery after PB (FRAP) over 16 min (n ≥ 9 animals per group). *p<0.01, ****p<0.0001 using an extra sum-of-squares *F*-test with a Bonferroni correction. (**F**) 20 s representative kymographs of GLR-1::GFP movement in AVA neurite in controls (strain: FJH 18) and *mcu-1(lf)* (strain: FJH 576) with or without Ru360 pretreatment. Time is represented on the y-axis and distance on the x-axis. (**G**) Total transport events quantified from kymographs in all conditions (n > 10 animals per group). (**H**) Representative traces of $\Delta F/F_{min}$ of cytoplasmic GCaMP6f following optical stimulation. (**I**) The maximum $\Delta F/F_{min}$ of cytoplasmic GCaMP6f events and (**J**) normalized total GCaMP6f activity during a 1.5 min recording with optical activation of AVA neurons every 30 s (n ≥ 10 animals per group). All scale bars = 5 μm. Data is represented as mean ± s.e.m.; n.s, not significant, ****p<0.0001 compared to controls or indicated experimental group using a one-way ANOVA with a Dunnett's test. Source data is available at https://doi.org/10.5061/dryad.0gb5mkm71.

The online version of this article includes the following video and figure supplement(s) for figure 2:

**Figure supplement 1.** Additional analysis of SEP::GLR-1 fluorescence and fluorescence recovery after photobleaching (FRAP), and velocity analysis of GLR-1 transport.

**Figure 2—video 1.** GLR-1::GFP transport in AVA neurons.

https://elifesciences.org/articles/92376/figures#fig2video1

---

(*Figure 2—figure supplement 1A*). Next, we used fluorescence recovery after photobleaching (FRAP) of SEP::GLR-1 to measure the rate of GLR-1 recruitment to the synaptic membrane. SEP will only fluoresce when GLR-1 is positioned at the plasma membrane and is quenched while in transport vesicles or synaptic endosomes (see *Appendix 2—figure 1A*). In addition, our FRAP protocol (see 'Materials and methods' for details) involves photobleaching a ~40–60 μm portion of the neurite proximally and distally to the imaging region that is intended to limit the influence of GLR-1 lateral diffusion in the membrane on fluorescence recovery. Thus, the relative recovery of SEP fluorescence (%FRAP rate) in a photobleached neurite is representative of GLR-1 that has been exocytosed to the membrane and the rate of GLR-1 recycled via endocytosis. The rate of SEP fluorescence recovery (without individual normalization; see 'Materials and methods' for analysis details) was increased more than twofold in *mcu-1(lf)* and slightly increased following Ru360 treatment (*Figure 2B and C*). When the fluorescence at each timepoint after photobleaching is normalized to the fluorescence before photobleaching, the %FRAP is unchanged between experimental groups (*Figure 2—figure supplement 1B*). Taken together, these analyses show that loss or inhibition of MCU-1 leads to excessive recruitment of GLR-1 to synapses but proportional to the amount of GLR-1 at synapses.

The recruitment of GLR-1 to the synaptic membrane depends on the local GLR-1 reserves in synaptic endosomes (*Gutiérrez et al., 2021*). Resupplying of these local receptor pools occurs when GLR-1-containing transport vesicles are delivered to endosomes or other local reserves (*Petrini et al., 2009*). The delivery rate of new GLR-1 can be measured by FRAP of GLR-1::GFP (see *Appendix 2—figure 1B*). In *mcu-1(lf)*, the rate of GLR-1::GFP FRAP was decreased compared to controls but slightly increased in Ru360-treated animals (*Figure 2D and E*). Synaptic delivery and exocytosis of GLR-1 are dependent upon the transport of GLR-1-containing vesicles by molecular motors from the cell body where GLR-1 is predominantly synthesized. So, to better understand our results above (*Figure 2C and E*), we analyzed GLR-1 transport in *mcu-1(lf)* and with Ru360 treatment. To do this, we visualized individual GLR-1::GFP transport by photobleaching a section (~40 μm) of the AVA neurites (*Figure 2—video 1*) as previously described (*Hoerndli et al., 2022*; *Doser et al., 2020*). Interestingly, we found that both *mcu-1(lf)* and Ru360 treatment decreased the amount of GLR-1 transport by ~50% (*Figure 2F and G*). Ru360 treatment of *mcu-1(lf)* did not further decrease the amount of GLR-1 transport compared to *mcu-1(lf)* alone. Mitochondrial matrix $Ca^{2+}$ regulates oxidative phosphorylation via several mechanisms, so *mcu-1(lf)* and/or Ru360 treatment could reduce GLR-1 transport indirectly by decreasing ATP production. The processivity and velocity of molecular motor movement are highly coupled to ATP availability (*Schnitzer et al., 2000*) but the velocity of GLR-1 transport was comparable between controls and *mcu-1(lf)* or with Ru360 (*Figure 2—figure supplement 1C*). This suggests that ATP availability is relatively unchanged by loss or inhibition of MCU-1 or that basal rates of ATP production are sufficient to support normal transport velocities. Together, these results suggest that mitochondrial $Ca^{2+}$ uptake differentially regulates GLR-1 transport out of the cell body and synaptic recruitment of GLR-1.

Previous work has shown that cytoplasmic $Ca^{2+}$ signaling regulates transport and synaptic localization of GLR-1 (*Hangen et al., 2018*; *Hoerndli et al., 2015*; *Doser et al., 2020*), so we tested if decreased mitochondrial $Ca^{2+}$ uptake alters the amplitude or duration of cytoplasmic $Ca^{2+}$ transients

in dendrites following neuronal activation since this would impact downstream $Ca^{2+}$ signaling and synaptic recruitment of GLR-1. We expressed ChRimson and the cytoplasmic $Ca^{2+}$ indicator GCaMP6f in the AVA neurons in *mcu-1(lf)* and control animals. This approach bypasses activation by presynaptic inputs allowing direct activation of the AVA interneurons. We simultaneously optically activated the AVA neurons and recorded GCaMP6f fluorescence in *mcu-1(lf)* and Ru360-treated controls in the same dendritic region of the AVA neurons where GLR-1 transport and FRAP were analyzed. There were no significant changes in cytoplasmic $Ca^{2+}$ transients in dendrites following AVA activation with ChRimson between *mcu-1(lf)* or with Ru360 treatment compared to controls (*Figure 2H–J*), suggesting that loss or inhibition of MCU-1 does not drastically alter activity-dependent cytoplasmic $Ca^{2+}$ influx or the duration of a $Ca^{2+}$ event in dendrites. In other words, the loss or inhibition of MCU-1 does not seem to impact synaptic recruitment of GLR-1 by indirectly modulating cytoplasmic $Ca^{2+}$ signaling.

## Neuronal excitation upregulates mitoROS signaling

Our previous work has shown that ROS regulate transport and synaptic delivery of GLR-1 (*Doser et al., 2020*; *Doser and Hoerndli, 2022*). To further address the mechanism by which $Ca^{2+}$ influx by MCU-1 modulates GLR-1, we tested if activity-dependent $Ca^{2+}$ uptake regulates mitoROS production. To do this, we stimulated the AVA neuron with ChRimson using the same optical activation that initiated mitochondrial $Ca^{2+}$ uptake (*Figure 1*). Then, we measured ROS levels at dendritic mitochondria using a genetically encoded ratiometric ROS sensor that was localized to the outer mitochondrial membrane (mito-roGFP) (*Morgan et al., 2011*; *Figure 3A*). We found that the duration of repetitive AVA stimulation positively correlated with the mito-roGFP fluorescence ratio ($F_{ratio}$; 405/488 nm), indicating increased ROS following neuronal activation (*Figure 3B and C*). The $F_{ratio}$ was unchanged in controls that were not treated with Retinal, which is required for optical stimulation, and subjected to the light stimulation protocol. Similar to mitoGCaMP responses, we saw diversity among dendritic mitochondria in mito-roGFP $F_{ratios}$ following neuronal activation of the AVA neurons (*Figure 3—figure supplement 1A and B*). The frequency distribution of mito-roGFP $F_{ratios}$ of individual mitochondria without stimulation is unimodal (centered at 0.03) but becomes bimodal following 60 min of repetitive activation. One peak is slightly right shifted (centered at 0.05) and the other is strongly right shifted, corresponding to significantly higher mito-roGFP $F_{ratios}$ (centered at 0.09; *Figure 3—figure supplement 1A and B*). These results suggest that mitochondria within these neurites differentially respond to activity in terms of their ROS production.

So, does this activity-dependent upregulation of mitoROS production require $Ca^{2+}$ uptake through MCU-1? Expression of ChRimson and mito-roGFP in *mcu-1(lf)* revealed that the loss of MCU-1 prevented activity-induced increases in the mito-roGFP $F_{ratio}$ even after 60 min of repetitive optical activation (*Figure 3D and E*, *Figure 3—figure supplement 1C*). Pretreatment with Ru360 prior to optical activation similarly prevented the activity-induced increase in mito-roGFP $F_{ratio}$ (*Figure 3F and G*, *Figure 3—figure supplement 1D*). In summary, both the acute pharmacological inhibition and genetic loss of MCU-1 prevented activity-dependent upregulation of mitoROS production. Since optical activation is artificial and does not rely on synaptic transmission, it is possible that mitoROS production is not upregulated by natural neuronal activation. To address this, we took advantage of the well-defined circuitry in *C. elegans* and designed an experiment to activate a subset of mechanosensory neurons that detect physical touch and vibration (*Schafer, 2015*) and provide excitatory input to AVA neurons. This involved repetitively activating presynaptic mechanosensory neurons with vibration caused by dropping culture plates containing freely behaving worms from a short distance (~5 cm) onto the bench top every 30 s for a duration of 5 or 10 min. Then, worms were mounted for imaging to assess the $F_{ratio}$ of mito-roGFP. The mito-roGFP $F_{ratio}$ was slightly increased following 5 min and significantly increased by 10 min of repetitive mechano-stimulation (*Figure 4A and B*), indicating that mitoROS production is also increased by native means of neuronal activation.

## Using the photosensitizer KillerRed for artificial ROS production at dendritic mitochondria

We next wanted to address the possible role of ROS production at dendritic mitochondria in regulating the multistep process required for synaptic recruitment of GLR-1 in a cell-specific manner independent of mitochondrial $Ca^{2+}$ handling. To this end, we expressed the photosensitizer KillerRed that produces ROS upon photoactivation (PA) with green light (*Bulina et al., 2006*). In addition, we localized

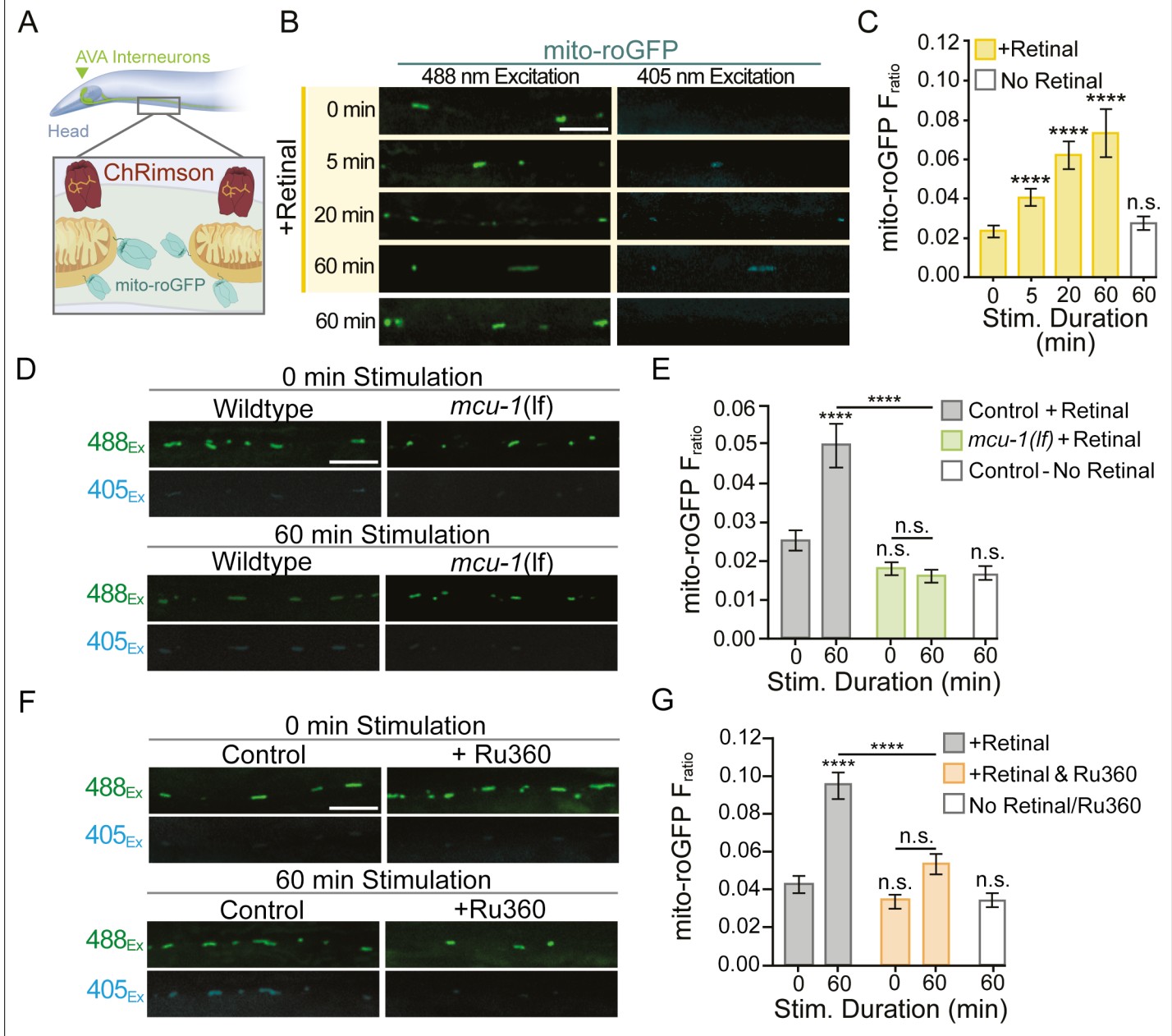

**Figure 3.** Mitochondrial reactive oxygen species (mitoROS) production is upregulated by neuronal activity and dependent on mitochondrial Ca²⁺ uptake via MCU-1. (**A**) Illustration showing transgenic expression and subcellular localization of ChRimson and mito-roGFP in the AVA neurons. (**B**) Representative images of mito-roGFP fluorescence in a single Z-plane when excited with 488 nm or 405 nm light following optogenetic stimulation with or without all-trans-Retinal (strain: FJH 402). (**C**) Mito-roGFP fluorescence ratio (405/488 nm) following 0, 5, 20, or 60 min of repetitive optical stimulation (40 μW/mm² at 33.3 mHz) with Retinal pretreatment and 60 min of repetitive optical stimulation without Retinal pretreatment (n > 30 mitochondria from eight animals per group). (**D, F**) Representative images of mito-roGFP fluorescence in a single Z-plane when excited by 488 nm or 405 nm light following 0 or 60 min of repetitive optical stimulation with Retinal pretreatment. (**E**) Mito-roGFP F_{ratio} following 0 or 60 min of repetitive optical stimulation in *mcu-1(lf)* (strain: FJH 706) and controls (strain: FJH 402), as well as non-Retinal-treated controls that underwent 60 min of stimulation (n ≥ 32 mitochondria from eight animals per group). Statistical comparisons are between groups and the 0 min control unless indicated by horizontal bar. (**G**) Mito-roGFP F_{ratio} at 0 and 60 min following repeated optical stimulation with or without Ru360 treatment (n ≥ 38 mitochondria from eight animals per group; strain FJH 402). All scale bars = 5 μm. Data is represented as mean ± s.e.m.; n.s., not significant, ****p<0.0001 compared to controls or indicated experimental group using a one-way ANOVA with a Dunnett's test. Source data is available at https://doi.org/10.5061/dryad.0gb5mkm71.

The online version of this article includes the following figure supplement(s) for figure 3:

**Figure supplement 1.** Additional analysis of mito-roGFP reveals differential activity-induced reactive oxygen species (ROS) production at dendritic mitochondria.

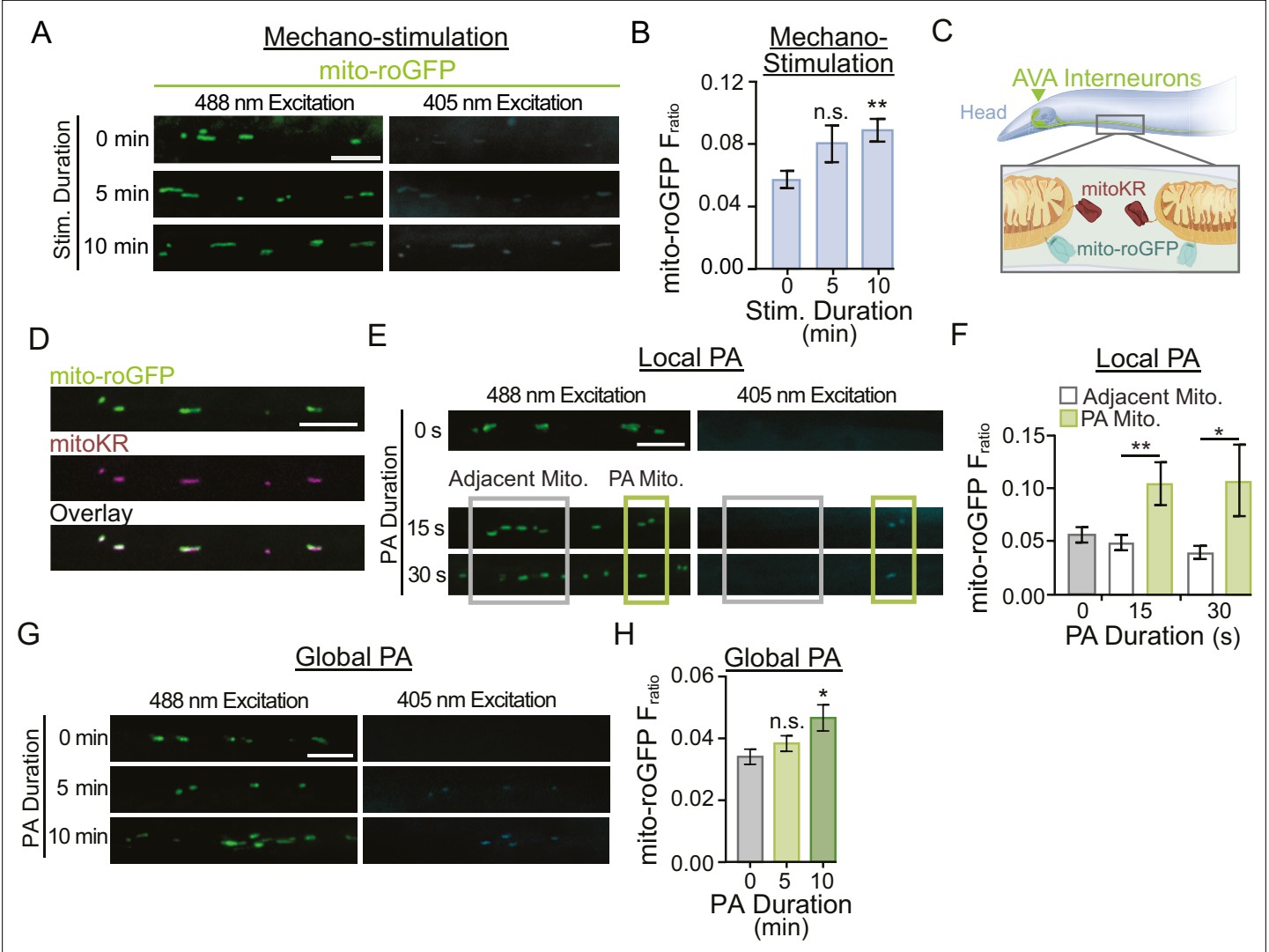

**Figure 4.** Photoactivation (PA) of mitochondria localized KillerRed results in physiological elevations in mitochondrial reactive oxygen species (mitoROS). (**A**) Representative images of mito-roGFP fluorescence in a single Z-plane when excited with 488 nm or 405 nm light following 0, 5, or 10 min of repetitive mechano-stimulation (strain: FJH 402). (**B**) Quantification of the average mito-roGFP $F_{ratio}$ following 0, 5, or 10 min of repetitive mechano-stimulation (n > 50 mitochondria from eight animals per condition). (**C**) Illustration depicting subcellular localization of mitoKR and mito-roGFP within the AVA neurites. (**D**) Representative fluorescent images demonstrating the co-localization of mitoKR and mito-roGFP within the AVA neurite (strain: FJH 416). (**E**) Representative fluorescent images of mito-roGFP when excited by 405 nm or 488 nm light following 0, 15, or 30 s of PA directed at 1–3 mitochondria (green box). Localization of PA was considered to be spatially specific enough that neighboring mitochondria (gray box) were not exposed to the PA stimulus. (**F**) The mito-roGFP $F_{ratio}$ in mitochondria that were (green bars; n = 8 mitochondria from eight worms per group) or were not (white bars; n > 15 mitochondria from eight worms) targeted for PA as well as in worms without any additional optical activation (gray bars). n > 20 mitochondria from eight worms per group; *p<0.05, **p<0.005 using a paired *t*-test. No significant difference between the no light controls and the neighboring mitochondria (one-way ANOVA with Dunnett's test). (**G**) Representative fluorescent images of mito-roGFP excited by 488 nm or 405 nm light with 0, 5, or 10 min of consistent light (567 nm; 0.025 mW/mm²) for global PA of mitoKR. (**H**) Quantification of mito-roGFP fluorescence ratio ($F_{ratio}$, Ex405/Ex488nm) for each group (n > 32 mitochondria from eight animals per group). All scale bars = 5 µm. Data is represented as mean ± s.e.m.; *p<0.05, n.s., not significant using a one-way ANOVA with a Dunnett's test. Source data is available at https://doi.org/10.5061/dryad.0gb5mkm71.

KillerRed to mitochondria (mitoKR) by anchoring it to the outer mitochondrial membrane with the localization tag TOMM20 (*Braeckman et al., 2016*). First, we co-expressed mitoKR with mito-roGFP (*Figure 4C and D*) for optimization of a PA protocol that would artificially induce elevations in ROS levels (within the physiological range) at a subset of synapses (local, *Figure 4E and F*) or throughout the AVA neuron (global, *Figure 4G and H*). To test our local PA protocol, we used a microscopy setup that was equipped for targeted illumination (see 'Materials and methods') allowing us to direct a

green LED to a small portion (~10 μm) of the AVA neurites containing 1–3 mitochondria for 15 or 30 s (*Figure 4E*). The mito-roGFP $F_{ratio}$ was increased in the mitochondria that were targeted for 15 or 30 s of PA when compared to non-activated controls as well as neighboring mitochondria not targeted for PA (*Figure 4F*). Local PA of mitoKR increased the mito-roGFP $F_{ratio}$ by 2×, which is comparable to the effect of short-term AVA activation by ChRimson (*Figure 3C*) and mechano-stimulation (*Figure 4A and B*) on mitoROS production. In addition, local PA of mitoKR had no effect on the amount of GLR-1 transport (data not shown) or on GLR-1 transport velocity (*Figure 5—figure supplement 1A*). Both of these processes rely on intact microtubules and normal microtubule dynamics that are sensitive to prolonged elevations in ROS (*Doser et al., 2020*; *Wilson and González-Billault, 2015*; *Debattisti et al., 2017*) and oxidative stress (*Goldblum et al., 2021*; *Drum et al., 2016*; *Fang et al., 2012*). In other words, local PA of mitoKR results in *physiological* elevations in mitoROS production.

Secondly, we optimized a protocol to modestly increase ROS production at mitochondria throughout AVA interneurons. More specifically, whole-cell PA of mitoKR was achieved by illuminating freely behaving worms for 5 or 10 min. We used mito-roGFP to measure the resultant ROS increase at mito-chondria from whole-cell PA and observed a slight increase in the average mito-roGFP $F_{ratio}$ after 5 min of whole-cell PA and a significant increase in the $F_{ratio}$ following a 10 min whole-cell PA (*Figure 4G and H*). Although not significantly increased from the unstimulated control, the 5 min PA increased the $F_{ratio}$ of mito-roGFP to 0.4, which is similar to the mito-roGFP $F_{ratio}$ following 5 min of repetitive ChRimson activation (*Figure 3C*) or mechano-stimulation of AVA (*Figure 4A and B*). Therefore, we chose to do subsequent experiments using a whole-cell PA duration of 5 min. Finally, this global mitoKR activation protocol did not affect overall GLR-1 transport velocity (*Figure 5—figure supplement 1B*), further supporting our choice of these conditions as relevant for signaling but non-toxic.

## Mitochondrial ROS signaling regulates synaptic recruitment of GLR-1

Once we established non-toxic conditions for local (2–3 mitochondria) and global (entire AVA neuron) mitoKR activation, we proceeded to test the effect of cell-specific and subcellular mitoROS signaling on synaptic GLR-1 recruitment. First, we used our local mitoKR protocol (15 s) to activate 2–3 mito-chondria prior to assessing synaptic recruitment of GLR-1 via FRAP of SEP::GLR-1 (*Figure 5A*). These experiments required the generation of new transgenic animals expressing SEP::GLR-1 in AVA with (strain: FJH 582) and without mitoKR (strain: FJH 635; see Appendix 1—key resources table). Inter-estingly, local PA dramatically decreased SEP::GLR-1 FRAP in mitoKR-expressing worms compared to controls (*Figure 5B and C*). The FRAP rate of non-activated mitoKR worms was significantly decreased compared to controls, but to a lesser extent than with PA (*Figure 5—figure supplement 1C*). This is likely due to activation of mitoKR during imaging of SEP fluorescence. This dramatic downregulation of GLR-1 synaptic recruitment due to localized artificial mitoROS production could be caused by altered delivery of GLR-1-containing transport vesicles. When we assessed GLR-1 delivery via FRAP of GLR-1::GFP following local PA of mitoKR, we observed that PA of mitoKR decreased the rate of GLR-1::GFP FRAP in worms expressing mitoKR in comparison to controls lacking mitoKR (*Figure 5D and E*), as well as mitoKR-expressing animals without PA (*Figure 5—figure supplement 1D*). These results suggest that the delivery and retention of GLR-1 to synaptic sites are negatively regulated by local mitoROS production.

We speculated that ROS production by mitoKR could also impact transport of GLR-1 in a similar fashion to global ROS elevations shown previously (*Doser et al., 2020*). To test this hypothesis, we subjected animals to our global mitoKR activation protocol prior to imaging GLR-1 transport (*Figure 5F*) and found that cell-wide PA of mitoKR reduces the number of transport events (*Figure 5G and H*). These results coincide with our previous work showing that systemic elevations in ROS decrease export of GLR-1 out of the cell body (*Doser et al., 2020*) and suggest that the mitochondria are a major source of the ROS involved in this regulation.

In summary, our results demonstrate that dendritic mitochondria take up $Ca^{2+}$ in response to neuronal activity, leading to an upregulation in ROS production at mitochondria. We also show that cell-specific ROS production at mitochondria and loss or inhibition of MCU had opposite effects on GLR-1 recruitment in AVA neurites (*Figures 2C and 5C*), so we hypothesized that local $Ca^{2+}$ uptake by mitochondria and mitoROS production regulate the amount of GLR-1 at the plasma membrane through the same signaling pathway. To test this, we subjected control or mitoKR-expressing worms to an acute Ru360 treatment, mounted them for imaging, and photoactivated a region of the AVA

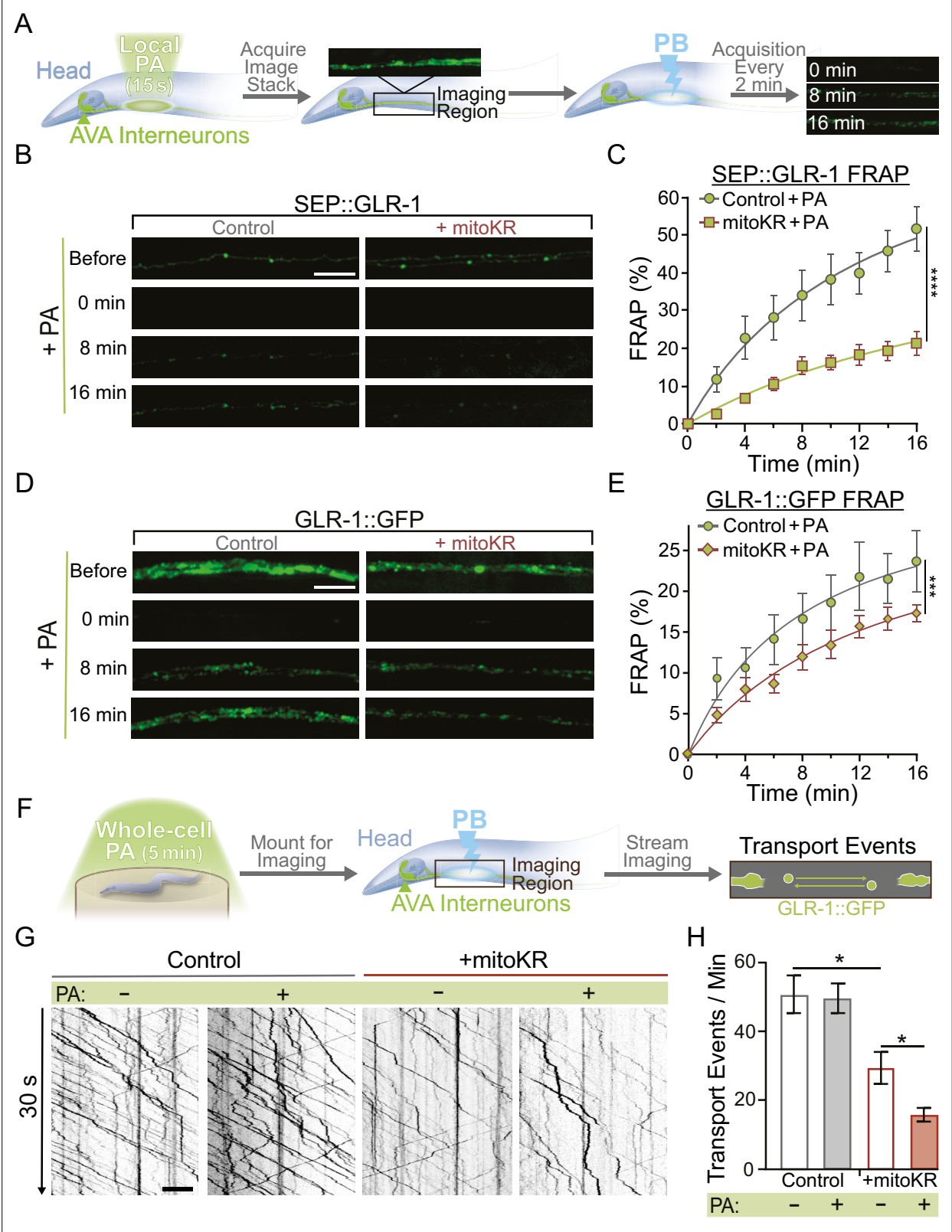

**Figure 5.** Mitochondrial reactive oxygen species (mitoROS) downregulates the recruitment of GLR-1 to synapses. (**A**) Diagram of experimental procedure followed for (**B–D**) (see 'Materials and methods'). (**B, D**) Representative images of (**B**) SEP::GLR-1 (strains: FJH 635 and FJH 582) or (**D**) GLR-1::GFP (strains: FJH 18 and FJH 555) fluorescence before, immediately after, 8, and 16 min after local photoactivation (PA) and photobleach (PB). (**C, E**) Percent SEP (**C**) or GFP (**E**) fluorescence recovery after PB (FRAP) over 16 min after local PA and PB (n ≥ 7 animals per group). ***p<0.0005,

*Figure 5 continued on next page*

*Figure 5 continued*

****p<0.0001 using an extra sum-of-squares *F*-test with a Bonferroni correction. (**F**) Diagram of experimental procedure followed for (**G, H**) (see 'Materials and methods'). (**G**) 30 s representative kymographs of GLR-1::GFP movement in the AVA with or without global PA. (**H**) Total number of transport events per minute quantified from 50-s-long kymographs (n = 8 animals per +mitoKR group, and n = 4 per control group). All scale bars = 5 μm. Data is represented as mean ± s.e.m.; *p<0.05 compared to controls or indicated experimental group using a one-way ANOVA. Source data is available at https://doi.org/10.5061/dryad.0gb5mkm71.

The online version of this article includes the following figure supplement(s) for figure 5:

**Figure supplement 1.** Post hoc velocity analyses of GLR-1 transport with mitoKR activation and non-photoactivation (non-PA) fluorescence recovery after photobleaching (FRAP) controls.

neurites prior to carrying out the FRAP protocol for SEP::GLR-1 (*Figure 6A*). This technique allowed us to acutely bypass mitochondrial $Ca^{2+}$ uptake and artificially induce ROS production at dendritic mitochondria in order to test if mitoROS is sufficient to downregulate synaptic recruitment of GLR-1 in the AVA neurites. In this experiment, Ru360 treatment increased SEP::GLR-1 %FRAP. This result is inconsistent with the effect of Ru360 on the %FRAP of SEP::GLR-1 presented in *Figure 2—figure supplement 1B*, but we speculate that this discrepancy may be due to lower basal expression of SEP::GLR-1 in these strains than those used previously (strains FJH 314 and FJH 638 used in *Figure 2* and *Figure 2—figure supplement 1B*; data not shown). Local PA of mitoKR decreased the recovery rate, and when combined with Ru360 treatment, the %FRAP of SEP::GLR-1 was slightly delayed, but the relative fluorescence recovery after 16 min post-photobleach was nearly identical to local PA of mitoKR alone (*Figure 6B and C*). Interestingly, Ru360 treatment of mitoKR-expressing worms without PA had a %FRAP rate that was comparable to the non-activated, untreated mitoKR group (*Figure 6—figure supplement 1A*). Since artificial mitoROS production was able to occlude the effect of Ru360 on SEP::GLR-1 FRAP, these results support that mitoROS is necessary and sufficient for downregulating recruitment of GLR-1 to synapses. Contrary to synaptic GLR-1 recruitment, somatic export of GLR-1 is paradoxically reduced by both artificial mitoROS production and inhibition of MCU-1. To test if mitoROS and mitochondrial $Ca^{2+}$ uptake regulate GLR-1 transport out of the cell body via the same mechanism, we combined acute Ru360 treatment with 5 min of whole-cell PA of mitoKR prior to imaging GLR-1 transport (*Figure 6D*). Both acute Ru360 treatment and whole-cell PA of mitoKR decreased the number of GLR-1 transport events to a similar extent (*Figures 2G and 5H*). When combined, the amount of GLR-1 transport was significantly decreased compared to Ru360 treatment alone and modestly decreased compared to mitoKR activation (*Figure 6E and F*). Ru360 treatment of mitoKR-expressing worms in the absence of PA had no additional effect on the amount of GLR-1 transport compared to untreated mitoKR-expressing worms (*Figure 6—figure supplement 1B*). The compounding effect of mitoROS production and decreased mitochondrial $Ca^{2+}$ uptake indicates that mitoROS signaling and mitochondrial $Ca^{2+}$ uptake modulate GLR-1 transport via parallel regulatory pathways. This contrasts our observations of a $Ca^{2+}$-dependent mitoROS signaling mechanism in the regulation of GLR-1 recruitment to synapses (*Figure 5D*) and suggests that mitochondrial activation and signaling vary based on subcellular location. Taken altogether, our results reveal a physiological mitoROS signaling mechanism that is initiated by activity-dependent $Ca^{2+}$ uptake and downregulates GLR-1 recruitment to synapses.

## Discussion

Taken together, our experimental results outline a possible novel activity-dependent mitochondrial signaling mechanism that negatively regulates excitatory synapse function. Our data suggest that mitochondrial $Ca^{2+}$ uptake and ROS production are involved in different regulatory mechanisms based on subcellular location and/or process. In the cell body, mitochondrial $Ca^{2+}$ uptake and ROS production influence GLR-1 export via parallel mechanisms (*Figure 7A*). Our results indicate that $Ca^{2+}$ influx through MCU-1 is required in the neuronal cell body for normal GLR-1 transport, suggesting that mitochondrial $Ca^{2+}$ positively regulates transport via unknown indirect signaling mechanisms (maybe ATP production, orange dashed arrow in *Figure 7A*). Independent of MCU-1 function, mitoROS downregulates somatic export of GLR-1 (red dashed inhibition arrow in *Figure 7A*), and as suggested by our previous work, this probably occurs by redox regulation of proteins involved in this process (*Hoerndli et al., 2015*; *Doser et al., 2020*). At postsynaptic sites, mitochondrial $Ca^{2+}$ uptake and ROS

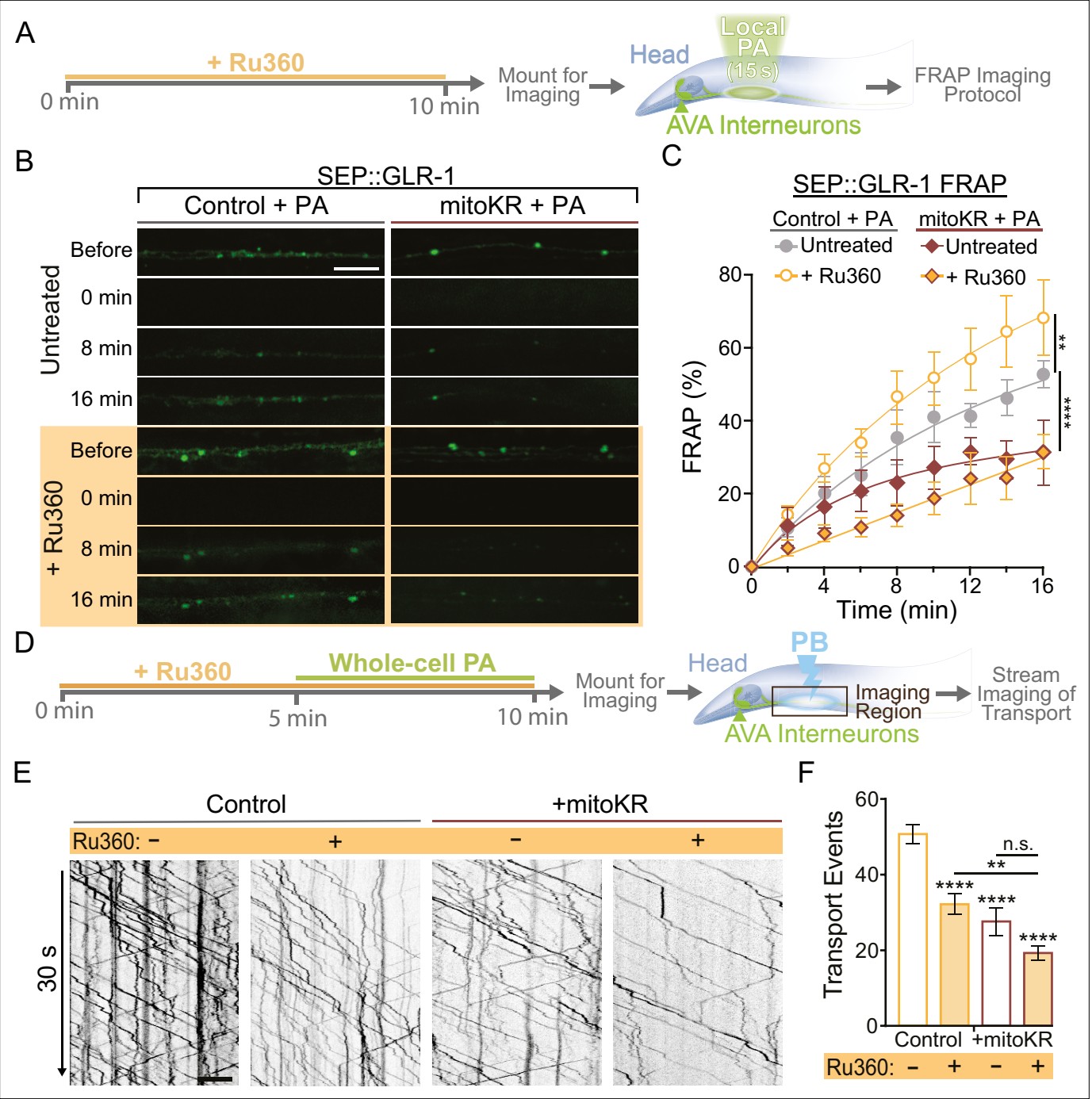

**Figure 6.** Regulation of synaptic recruitment of GLR-1 by mitochondrial reactive oxygen species (mitoROS) requires $Ca^{2+}$ uptake via MCU-1. (**A**) Diagram of experimental procedure in (**B, C**) (see 'Materials and methods'). (**B**) Representative images of SEP fluorescence prior to, immediately after, and at 8 and 16 min post photobleach (PB). (**C**) Percent SEP fluorescence recovery after photobleaching (FRAP) throughout 16 min post PB in controls (strain: FJH 635) or mitoKR-expressing animals (strain: FJH 582) ± Ru360 treatment with photoactivation (PA) (n = 6 animals per group). **$p < 0.005$, ****$p < 0.0001$ using an extra sum-of-squares $F$-test with a Bonferroni correction. (**D**) Diagram of experimental procedure for (**E, F**) (see 'Materials and methods'). (**E**) 30-s-long representative kymographs of GLR-1 transport in controls (strain: FJH 18) or mitoKR-expressing animals (strain: FJH 555) ± Ru360 treatment with PA. (**F**) Total number of transport events quantified from 50-s-long kymographs (n ≥ 10 animals per group). All scale bars = 5 μm. Data is represented as mean ± s.e.m.; n.s., not significant, **$p < 0.005$, ****$p < 0.0001$ compared to controls or indicated experimental group using a one-way ANOVA. Source data is available at https://doi.org/10.5061/dryad.0gb5mkm71.

The online version of this article includes the following figure supplement(s) for figure 6:

**Figure supplement 1.** Additional non-photoactivation (non-PA) SEP fluorescence recovery after photobleaching (FRAP) and transport controls.

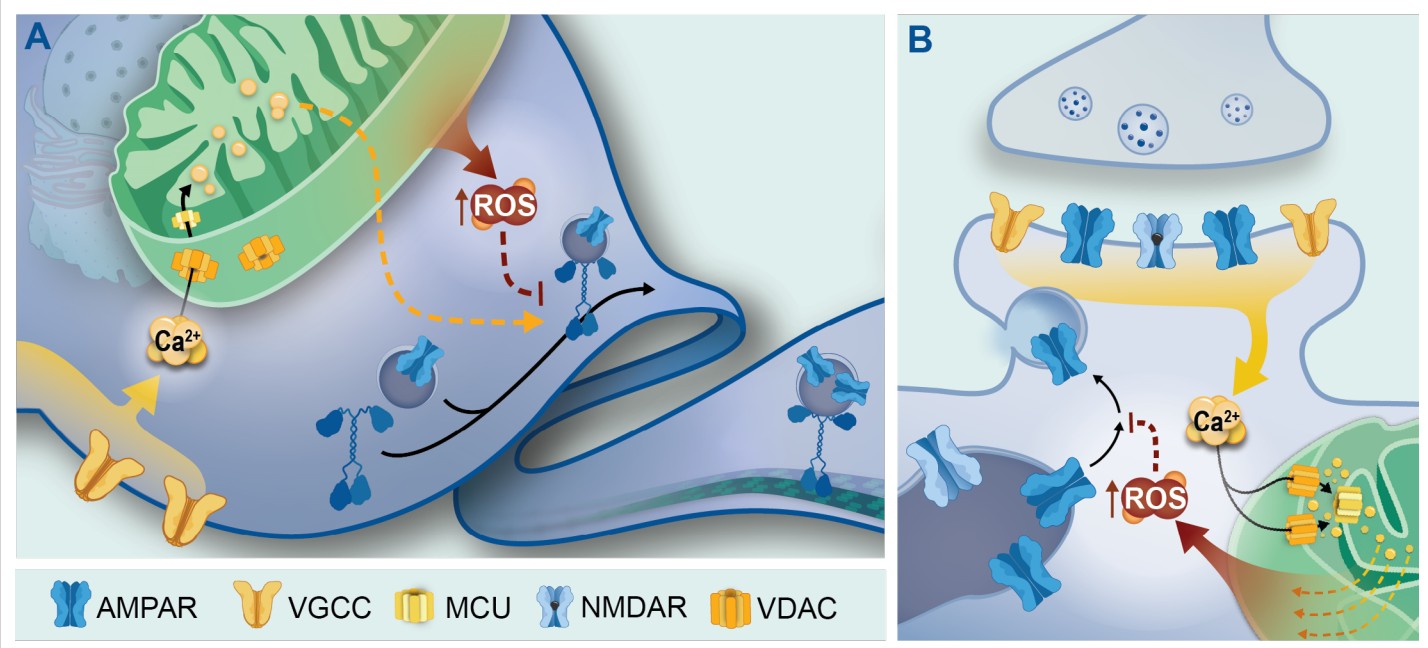

**Figure 7.** Proposed model. In neurons, increased cytoplasmic Ca$^{2+}$ due to activity-dependent opening of AMPARs, NMDARs, and voltage-gated calcium channels (VGCCs) results in mitochondrial Ca$^{2+}$ uptake via voltage-dependent anion channels (VDACs) at the outer mitochondrial membrane and further entry into the mitochondrial matrix via MCU. Once in the matrix, Ca$^{2+}$ can directly and indirectly upregulate mitochondrial respiration from which reactive oxygen species (ROS) is a by-product. The increased ROS can escape into the cytoplasm in the form of H$_2$O$_2$ and contribute to ROS signaling. This research points toward differential roles for and interactions between MCU and mitochondrial ROS (mitoROS) in regulating the subcellular trafficking of GLR-1. The results presented here indicate that (**A**) in the neuronal soma, where GLR-1 is synthesized and then exported, MCU-1 function indirectly promotes (dashed orange arrow) GLR-1 export, whereas mitoROS indirectly inhibits (dashed red inhibition arrow) it by acting on undetermined proteins. Alternatively, our data suggest that at postsynaptic sites, (**B**) ROS signaling resulting from Ca$^{2+}$ uptake via MCU may target and modulate the function of undetermined proteins (dashed red line) that regulate the recruitment of AMPARs from transport vesicles or intracellular reserves (i.e., synaptic endosomes, left organelle) to the synaptic membrane and/or their synaptic retention.

production regulate the recruitment (and perhaps recycling) of GLR-1 to the synaptic membrane via a linear signaling mechanism (*Figure 7B*). We speculate that neuronal activation leads to mitochondrial Ca$^{2+}$ uptake via MCU-1, causing an increase in mitoROS that indirectly downregulates synaptic recruitment of AMPARs (*Figure 7B*). The effect of mitoROS signaling on AMPAR recruitment to synapses appears to be due to the compounding effect of decreased transport out of the cell body, synaptic delivery, as well as exocytosis of AMPARs to the synaptic membrane (*Figure 5*). This negative regulation by mitoROS may be a homeostatic mechanism that is important for the prevention of excessive synaptic strengthening and the excitotoxicity that could result without this regulatory mechanism. This model (*Figure 7*) is in alignment with our overall experimental results. However, further investigation about local GLR-1 trafficking in the context of our proposed model, and the molecular players involved, will be required to test this mechanism.

## Mitochondrial calcium handling in synaptic function and plasticity

Buffering of cytoplasmic Ca$^{2+}$ by mitochondria is thought to shape the spatiotemporal dynamics of Ca$^{2+}$ signaling and upregulate mitochondrial output to meet energy demands (*Duchen, 2000*). Fine regulation of synaptic Ca$^{2+}$ is particularly important because synaptic function and plasticity rely on a multitude of Ca$^{2+}$-dependent signaling pathways that are all sensitive to the amplitude and duration of elevated Ca$^{2+}$ (*Nakahata and Yasuda, 2018*). It is known that Ca$^{2+}$ handling by presynaptic mitochondria modulates various presynaptic mechanisms central to synaptic transmission and plasticity, including synaptic vesicle recycling (*Billups and Forsythe, 2002*; *Marland et al., 2016*) and release probability (*Ashrafi et al., 2020*; *Sun et al., 2013*; *Lee et al., 2007*; *Devine et al., 2022*). Electron microscopy has revealed that mitochondria in the pre- and postsynaptic compartments of excitatory synapses differ in both size and electron density (*Freeman et al., 2017*), hinting that postsynaptic

mitochondrial specialization is different from their presynaptic counterparts. However, only a few recently published studies have investigated if and how mitochondrial $Ca^{2+}$ handling in dendrites regulates synaptic function or plasticity (*Groten and MacVicar, 2022*; *O'Hare et al., 2022*), and none have assessed the direct link between mitochondrial signaling and postsynaptic function in healthy neurons.

Postsynaptic plasticity mechanisms are also highly sensitive to the concentration and duration of elevated $Ca^{2+}$ (*Huganir and Nicoll, 2013*; *Citri and Malenka, 2008*), so $Ca^{2+}$ uptake by postsynaptic mitochondria could shape $Ca^{2+}$ events, and therefore synaptic transmission (*O'Hare et al., 2022*). The importance of postsynaptic mitochondria for synaptic function could also be inferred from the decreased presence of synaptic mitochondria in Alzheimer's and Parkinson's disease that is observed before synaptic dysfunction (*Sheng, 2014*). Interestingly, mitochondrial transport in neurites is regulated by relative $Ca^{2+}$ levels such that mitochondria deposition occurs at regions of high $Ca^{2+}$, such as at pre- and postsynaptic sites (*Sheng, 2014*). If mitochondrial $Ca^{2+}$ buffering truly contributes to cytoplasmic $Ca^{2+}$ signaling, then one would expect an increase in cytoplasmic $Ca^{2+}$ levels when mitochondrial $Ca^{2+}$ uptake is diminished. It has been shown that loss of MCU-1 increases the amplitude and/or duration of cytoplasmic $Ca^{2+}$ events in both invertebrate and vertebrate neurons (*Groten and MacVicar, 2022*; *Bisbach et al., 2020*; *Nichols et al., 2017*). However, we did not detect a significant change in activity-dependent $Ca^{2+}$ influx in the AVA neuron's cytoplasm due to loss or inhibition of MCU-1 (*Figure 2H–J*). This discrepancy may be due to GCaMP6f's high affinity for $Ca^{2+}$ occluding slight changes in cytoplasmic $Ca^{2+}$. Alternatively, mitochondrial $Ca^{2+}$ uptake in AVA neurons, and perhaps *C. elegans* neurons in general, may be less reliant on MCU-1 function. It is also important to note that in our hands the loss or pharmacological inhibition of MCU-1 did not completely abolish mitochondrial $Ca^{2+}$ uptake. However, our observations are consistent with previous studies in which MCU-1 was conditionally or completely knocked out (*Álvarez-Illera et al., 2020*; *Hamilton et al., 2018*).

In addition to the importance of mitochondrial $Ca^{2+}$ buffering for cytoplasmic signaling, there are many $Ca^{2+}$-dependent processes within mitochondria. First, mitochondrial $Ca^{2+}$ uptake can upregulate OXPHOS, and therefore ATP production, via several direct and indirect mechanisms (*Rossi et al., 2019*). For example, $Ca^{2+}$ binds to and modulates the activity of multiple tricarboxylic acid cycle enzymes (*Rizzuto et al., 2012*; *Giorgi et al., 2018*; *O'Hare et al., 2022*), which upregulates production of the OXPHOS substrates NADH and FAD2 to indirectly impact ATP and ROS production. $Ca^{2+}$ can also more directly upregulate OXPHOS by binding to components of the electron transport chain and ATP synthase (*Rizzuto et al., 2012*; *Giorgi et al., 2018*; *O'Hare et al., 2022*). It is possible that loss or inhibition of MCU-1 prevents activity-dependent upregulation of ATP that may indirectly impact endergonic mechanisms, including GLR-1 transport, delivery, and exocytosis (*Schnitzer and Block, 1997*; *Hanley, 2007*; *Araki et al., 2010*). However, our observations of upregulated GLR-1 delivery and exocytosis when MCU-1 is mutated or inhibited (*Figure 2C*) suggest that when mitochondrial $Ca^{2+}$ uptake is decreased, ATP levels remain sufficient for local GLR-1 trafficking. Secondly, since ROS are a by-product of OXPHOS, $Ca^{2+}$ uptake can upregulate ROS production via several $Ca^{2+}$-dependent mechanisms (*Görlach et al., 2015*). In fact, activity-induced mitoROS production via an MCU-1-dependent mechanism has been described in *C. elegans* in epidermal wound healing (*Xu and Chisholm, 2014*). There is also evidence from in vitro studies in various human cell lines that MCU-dependent mitoROS signaling occurs in pathophysiological contexts such as during inflammation or hypoxia (*Dong et al., 2017*). Lastly, mitochondrial $Ca^{2+}$ uptake appears to be central to the pathophysiological plasticity mechanism that underlies hyperalgesia (*Kim et al., 2011*). However, this work, in addition to these previous studies, prompts more questions than it answers regarding postsynaptic roles of $Ca^{2+}$-dependent mitoROS production.

## Regulation of AMPAR trafficking by mitochondrial ROS signaling

The characteristics of ROS production and methods of action make them a diverse messenger molecule in various cell types, especially in the brain where metabolic activity and antioxidant mechanisms are higher than that in other tissues (*Biswas et al., 2022*; *Vicente-Gutiérrez et al., 2021*). ROS signaling can be localized and compartmentalized due to the localization of ROS sources such as at the plasma membrane via NADPH oxidase or at mitochondria that is balanced by rapid cytoplasmic ROS scavenging (*Niemeyer et al., 2021*; *Sies, 2017*). This is estimated to limit ROS diffusion to

around 1 µm from its source (*Lim et al., 2015*). Reversible protein oxidation by ROS is reminiscent of phosphorylation in that it can regulate protein folding, activation, and interactions (*Miseta and Csutora, 2000*). Interestingly, the proportion of oxidizable protein residues is increased fourfold in mammals compared to prokaryotes, suggesting that ROS signaling may contribute to organismal complexity (*Go and Jones, 2013*).

Although mitochondria are regarded as the predominant source of ROS, there has been very little investigation of physiological mitoROS signaling in neurons in vivo. Recently, however, mitoROS production was shown to promote secretion of a neuropeptide from sensory neurons in *C. elegans*, which activates antioxidant mechanisms in distal tissues (*Jia and Sieburth, 2021*). There are also a few studies that demonstrate the functional relevance and versatility of mitoROS signaling in vertebrate neurons and their circuitry (*Bao et al., 2009*; *Accardi et al., 2014*). Our results support an important mitochondrial signaling role and suggest a mechanism in which activity-dependent mitoROS production can regulate AMPAR recruitment. A comprehensive understanding of this mechanism would require systematically analyzing how protein oxidation alters the functionality of key players that regulate AMPAR delivery and recruitment to synapses.

There are several oxidizable candidate proteins and signaling molecules that regulate synaptic recruitment of AMPARs in neurons. Two major components of the $Ca^{2+}$-signaling cascade that positively regulate AMPAR transport are calmodulin (CaM) and $Ca^{2+}$/CaM-dependent protein kinase II (CaMKII) (*Hangen et al., 2018*; *Hoerndli et al., 2015*; *Doser et al., 2020*), which are functionally regulated by oxidation. CaM has two conserved methionines, and when oxidized, the binding and activation of CaM to CaMKII are reduced (*Robison et al., 2007*). When CaMKII is in its active $Ca^{2+}$/CaM-bound conformation, oxidation of the regulatory domain enhances kinase activity (*Erickson et al., 2008*). Alternatively, when CaMKII is inactive, oxidation within the CaM binding domain prevents association of $Ca^{2+}$/CaM with CaMKII (*Konstantinidis et al., 2020*). At postsynaptic sites, recycling of AMPARs is regulated in a CaM/CaMKII-dependent manner, meaning redox modification of these proteins can also influence AMPAR exocytosis and endocytosis at synapses (*Bayer and Schulman, 2019*). Other proteins that regulate this process include protein kinase C (PKC) (*Boehm et al., 2006*) and the PDZ domain-containing scaffold protein interacting with C kinase 1 (PICK-1) (*Fiuza et al., 2017*). Activation of PKC following synaptic activation increases AMPAR insertion at synaptic membranes (*Ren et al., 2013*), whereas PICK-1 regulates AMPAR endocytosis (*Fiuza et al., 2017*). Interestingly, ROS signaling can bidirectionally modulate PKC activity (*Steinberg, 2015*) and oxidation of PICK-1 prevents its association with the synaptic membrane (*Shi et al., 2010*). Although the effect of PICK-1 oxidation on synaptic expression of AMPARs has not been characterized, there is evidence that this redox mechanism regulates glutamatergic transmission and is protective during oxidative stress (*Wang et al., 2015*). Thus, the current study opens the door to other questions regarding redox regulation of synaptic function and plasticity.

In contrast to the regulation of synaptic recruitment of AMPARs by mitoROS signaling (*Figure 6B and C*), we observed a compounding effect of MCU-1 inhibition and artificial mitoROS production on AMPAR export from the cell body (*Figure 6E and F*). These results suggest that AMPAR transport out of the cell body is regulated by mitochondrial $Ca^{2+}$ handling and mitoROS production via two parallel signaling pathways. Since somatic mitochondria are morphologically distinct from their dendritic and axonal counterparts (*Lee et al., 2018*), it is possible that they are functionally different as well. Altogether, these results open the door to questions regarding how functional diversity among mitochondria may allow mitochondrial signaling to differentially regulate signaling pathways based on subcellular location.

## Implications and conclusion

Synaptic diversity is thought to enhance the computing power of the nervous system allowing for complex behaviors, a broad range of emotional states, and nearly endless memory storage. Interestingly, the proteomes of synaptic and non-synaptic mitochondria suggest that synaptic diversity may be enhanced by their resident mitochondria (*Stauch et al., 2014*; *Graham et al., 2017*). The proteomes of synaptic mitochondria allow for specialized functions, including activity-dependent regulation of ATP production and discrete $Ca^{2+}$ handling abilities (*Faria-Pereira and Morais, 2022*; *Brown et al., 2006*). The functional significance of enhanced energy capability and $Ca^{2+}$ handling has been assessed for presynaptic mitochondria, but not in the context of postsynaptic sites. Here, we provide data

indicating that postsynaptic mitochondria are functionally diverse and play a novel signaling role in regulating postsynaptic function.

In conclusion, we present evidence for a novel role of mitochondria in regulating the number of AMPARs at the synaptic membrane. This study proposes a model in which $Ca^{2+}$ signaling regulates mitoROS production differentially at the soma and synapses, providing a means of negative regulation of synaptic excitability in a way that may be important for synaptic homeostasis and prevention of excitotoxicity. This role for ROS signaling challenges the long-held misconception that elevated ROS is only detrimental to cells causing dysfunction and death (*Sies and Jones, 2020*). Instead, mitoROS signaling acts as a physiological signal integrating synaptic function and mitochondrial output to link neuronal connectivity and metabolic capacity. Although additional studies are required to test and refine our working model, it opens the door to many new and impactful questions.

## Materials and methods

### Plasmid construction

See Appendix 1—key resources table for details on plasmids used in this study. Plasmids were created using In-Fusion Cloning (Takara Bio) or the Gateway recombination (Invitrogen) method. DNA primers were created using Takara Bio's online In-Fusion Primer Design Tool for In-Fusion Cloning and with the open-source ApE Plasmid Editor (M. Wayne Davis) for the Gateway recombination method.

### *C. elegans* strains

*C. elegans* strains were maintained under standard conditions (*Stiernagle, 2006*) (NGM with OP50 20°C). All animals used in the experiments were 1-day-old adult hermaphrodites that were selected 24 hr prior to the experiments at the L4 stage. Transgenic strains (see Appendix 1—key resources table) were created by microinjection (*Evans, 2006*) of *lin-15(n765ts)* worms with DNA mixes composed of the plasmids described in Appendix 1—key resources table. All DNA mixes included a plasmid containing *lin-15(+)* to allow for phenotypic rescue of transgenic strains (*Praitis and Maduro, 2011*). All strains used in optogenetic experiments were also mutant for the *lite-1* gene (allele: *ok530*) to limit off-target effects of our optical stimulation protocols due to LITE-1 (*Gong et al., 2016*). This protocol for the introduction of recombinant DNA into *C. elegans* has been approved by the National Institutes of Health Institutional Biosafety Committee (protocol no. 18-043B).

### Confocal microscopy

All imaging was done using a Yokogawa CSUX1 spinning disc incorporated into a confocal microscope (Olympus IX83) with 405, 488, and 561 nm diode lasers (100–150 mW each; Andor ILE Laser Combiner). Images were captured using an Andor iXon Ultra EMCCD (DU-867) camera and a ×100/1.40NA oil objective (Olympus). Devices were controlled remotely for image acquisition using MetaMorph 7.10.1 (Molecular Devices).

### In vivo imaging of the AVA neurites

One-day-old adult hermaphrodites were mounted for imaging by placing a single worm on an agar pad (10% agarose dissolved in M9 buffer) on a microscope slide with 1.6 µL of a solution containing equal measures of polystyrene beads (Polybead, Cat# 00876-15, Polysciences Inc) and 30 mM muscimol (Cat# 195336, MP Biomedicals). Once the muscimol slowed worm movement (~5 min), a coverslip was dropped onto the agar pad, physically restraining the worm. The worm's orientation was manually adjusted by sliding the coverslip to reorient the positioning of the AVA interneurons for imaging (*Doser et al., 2023*).

### Whole-cell neuronal stimulation with ChRimson

Worms from strains expressing ChRimson were picked at the L4s stage onto an NGM/OP50 plate coated with a 100 µM concentration of all-trans-Retinal (Sigma-Aldrich, Cat# R2500-25; diluted with M9 buffer). Worms were left overnight on Retinal plates before optical neuronal activation via an LED array (613 nm, CoolBase 7 LED module from LuxeonStar). ChRimson expression was verified in these strains behaviorally by testing light-induced reversals (data not shown). For ChRimson activation before mito-roGFP imaging, freely behaving 1-day-old adults were placed onto a fresh NGM/

OP50 plate 2 inches beneath a 613 nm LED array. LED intensity was adjusted at the beginning of each experiment to 40 µW/mm² using a custom potentiometer in combination with a digital optical power console (ThorLabs, PM100C) and photodiode sensor (ThorLabs, S170C). The pattern generator pulsed the LED for 1 s every 30 s (33.3 mHz) for 5–60 min before worms were mounted for imaging.

## Localized ChRimson activation

To activate ChRimson within discrete regions of the AVA neurons, the neurites were located using a ×100 objective, the co-expressed fluorescent reagents (i.e., mito-roGFP, GCaMP, or mitoGCaMP), and the 488 nm imaging laser. Briefly, a fluorescent image of the co-expressed reagent in a single Z-plane was acquired and a region mask was created on the AVA neurites. Then, the green LED (with a 605+20 nm filter; Chroma) from an LED illumination system (CoolLED pE300ultra) illuminated the masked region via projection through a Mosaic II digital mirror device (DMD; Andor Mosaic 3) controlled remotely using MetaMorph. LED intensity was adjusted to a total output of 5 µW using a digital optical power console (ThorLabs, PM100C) and microscope slide thermal sensor (ThorLabs, S175C). During the acquisition of an image stream, the master shutter of the DMD was controlled using MetaMorph's 'Trigger Components' function to illuminate the masked region for 3 s every 30 s.

## Ratiometric fluorescence imaging and analysis of mito-roGFP

Immediately after ChRimson or mechano-stimulation, worms were mounted for imaging in a 15 mM Muscimol solution. The AVA neurites containing roGFP+ mitochondria were located, and images were collected with a 500 ms exposure every 0.25 µm to capture a stack of images (5.25 µm) around the neurites. The 525 nm emission was imaged with 405 nm, then 488 nm illumination at each Z-plane. The average roGFP 525 nm fluorescence from 405 or 488 nm excitation was measured at individual mitochondria using MetaMorph's region measurement tool in a single Z-plane where the roGFP fluorescence due to 488 nm excitation was the highest. The average background fluorescence near each mitochondrion was also collected. The mitochondria region trace was copied to the fluorescence image collected with 405 nm excitation at the corresponding Z-plane, then roGFP and background fluorescence values were logged.

## Whole-cell mitoKR activation

Individual 1-day-old adults of transgenic strains (*csfEx168, csfEx195,* or *csfEx188*) containing pRD36 (*Pflp-18::TOMM20::KillerRed::let-858*) as determined by the absence of the multi-vulva phenotype were transferred onto a fresh NGM/OP50 culture plate and placed 2 inches below a 567 nm LED array (CoolBase 7 LED module from LuxeonStar). The light intensity was adjusted to 25 µW/mm² with our potentiometer, digital optical power console, and photodiode sensor (S130C). Worms were illuminated for 5 or 10 min before being immediately mounted for imaging.

## Local mitoKR activation

For localized photoactivation of mitoKR (TOMM20::KillerRed), the AVA neurites were located using a ×100 objective, the co-expressed fluorescent reagents (i.e., mito-roGFP, GLR-1::GFP, or SEP::GLR-1), and the 488 nm imaging laser. An image of mitoKR fluorescence in a single Z-plane was briefly acquired using a 100 ms exposure time and 561 nm imaging laser. Using this image, a region mask was created around a small region (100–300 µm²) containing mitoKR+ mitochondria. The green LED (with a 590+20 nm filter; Chroma) from our LED illumination system illuminated the masked region via projection of the green light through our DMD controlled using MetaMorph. LED intensity was adjusted to a total output of 10 µW using a digital optical power console (ThorLabs, PM100C) and photodiode sensor (ThorLabs, S130C). By remotely opening the DMD master shuttler, the masked region was illuminated for 15 s.

## Pharmacological inhibition of MCU-1 with Ru360

Ru360 (Sigma-Aldrich, Cat# 557440) was reconstituted in water at a concentration of 2 mM, then distributed into 15 µL aliquots (in light safe microcentrifuge tubes) and stored at 4°C. Immediately before treatment an Ru360 aliquot was diluted to 100 µM with M9 buffer. Then, 2–3 animals were placed on an NGM plate with OP50 and 200 µL of 100 µM Ru360 solution was pipetted onto the OP50 lawn where the animals resided, completely covering the lawn. Treatment was applied for 10 min,

after which the animal was removed to be used in the outlined imaging protocols. For long-term optogenetic experiments, animals were bathed in the Ru360 treatment for ~10 min before the Ru360-containing media naturally absorbed into the NGM/OP50 plate. The animals remained on this plate while undergoing the optical activation protocol for 5–60 min (see 'Whole-cell neuronal stimulation with ChRimson' and 'Whole-cell mitoKR activation').

### Transport imaging and analysis

All transport imaging was conducted on strains containing *akIs141* in the *glr-1* null background (*ky176*). The AVA neurites were located using the ×100 objective and a 488 nm excitation laser to visualize GFP fluorescence. A consistent Z-plane was held in focus for the entire imaging session using the continuous focus function of a Z drift compensator (Olympus, IX3-ZDC2) controlled remotely using MetaMorph. Then, a proximal section of the neurites was photobleached using a 3 W, 488 nm Coherent solid-state laser (Genesis MX MTM; 0.5 W output; 1 s pulse) directed to the region defined in MetaMorph using a Mosaic II digital mirror device (Andor Mosaic 3). Then, 30 s after photobleaching, an image stream was collected with the 488 nm excitation laser and a 100 ms exposure time. Meta-Morph's Kymograph tool was used to generate kymographs as previously reported (*Hoerndli et al., 2013*). Transport events were quantified by manually counting all transport events from the resultant kymographs. Instantaneous transport velocities were quantified from kymographs using the ImageJ plugin KymoAnalyzer (*Neumann et al., 2017*) as previously described (*Doser et al., 2020*).

### Fluorescence recovery after photobleaching (FRAP)

Strains expressing either GLR-1::GFP or SEP::GLR-1 were mounted for imaging as described above. Using the SEP or GFP fluorescence, a proximal region of the AVA neurites was localized. The stage position was memorized using MetaMorph's stage position memory function and the ideal Z-plane was set using the ZDC control dialogue. An image stack of SEP/GFP fluorescence was then acquired using the 488 nm excitation laser set to a 500 ms exposure. The Z-stack captures the entire width of the AVA process (21 Z-planes; 0.25 µm steps, ± 2.5 µm around the neurite). If photoactivation was required for experiment, the shutter for the CoolLED system (pE-300^ultra) was opened for the appropriate duration. Then, ~40 µm sections of the neurite proximal and distal to the imaging region were photobleached using the same photobleaching settings as described for GLR-1 transport imaging. Lastly, the imaging region (40–50 µm) was photobleached. Immediately following, an image stack of SEP/GFP fluorescence was acquired for the 0 min timepoint. Subsequent image stacks were acquired every 2 min out to 16 min. The resultant image stacks were processed and analyzed as previously described (*Doser et al., 2020*), with the exception of the SEP FRAP dataset in *Figure 2C*. The individual timepoints in this dataset were not normalized to the initial fluorescence value per animal because initial SEP::GLR-1 fluorescence was significantly higher in *mcu-1(lf)* (*Figure 2—figure supplement 1A*). Instead, the 0 min fluorescence values were subtracted from the raw fluorescence values for all subsequent timepoints. Analysis of the fluorescence before photobleaching was analyzed by creating a data file (.log) of fluorescence values along the bleached region of the AVA neurite using MetaMorph's linescan tool (line width = 20 pixels). The resultant output file was analyzed using a custom MATLAB (R2021a) script to obtain the average area of fluorescent puncta (area under the peak).

### Imaging of mitoGCaMP and cytoplasmic GCaMP

The AVA neurite was located and continuous autofocus was set as described above. Image streams (100 ms exposure) were collected with a 488 nm imaging laser (power = 0.1%; attenuation = 10). Localized ChRimson activation (see 'Localized ChRimson activation') was triggered every 30 s using MetaMorph's 'Trigger Components' feature starting 30 s after the start of the image stream. Imaging of mitoGCaMP fluorescence was continuous throughout the entire protocol covering all aspects of activation and rest.

The AVA neurite was located and continuous autofocus set as described above. Then, a 90 s image stream was collected with a 488 nm imaging laser (set to 0.1% power and an attenuation of 10) and a 250 ms exposure. Localized ChRimson activation (see 'Localized ChRimson activation') was triggered every 30 s using MetaMorph's 'Trigger Components' feature starting 15 s after the start of the stream acquisition. Imaging of GCaMP6f fluorescence was continuous throughout the optical activation protocol.

## Experimental design and statistical analyses

All relevant controls were included for each set of biological replicates and all datasets combine 2–5 replicates conducted on different days. Appropriate sample size for each experiment was based on previously published experiments (*Hoerndli et al., 2013*; *Doser et al., 2020*). A post hoc Pearson's *R* correlation test was conducted for each dataset to ensure a small effect size ($|r| < 0.3$). When manual quantification was required (i.e., for quantification of transport events from kymographs), the dataset was blinded to the genotype and experimental condition. Outliers were removed from datasets using the ROUT method ($Q = 1\%$). For FRAP datasets, animals were excluded if 50% or more of the time-points were considered outliers. Experimental groups were considered significantly different if their comparison using a Student's *t*-test (for comparing two groups) or one-way ANOVA with correction for multiple comparisons (Dunnett's or Bonferroni's; for comparisons >2) yielded a p-value<0.05. To compare the FRAP rate between conditions, we used an extra sum-of-squares *F*-test comparing the best-fit curve for each experimental group with a Bonferroni correction for multiple comparisons. Curves were considered different if a comparison yielded a p-value<0.01.

## Image and data presentation

All images were acquired under non-saturating conditions. Representative images were selected as they represent the average. Postprocessing was done following analysis as needed to visualize corresponding quantifications. Images processed for data representation were performed using Photoshop (2023), and all images in each figure panel were identically processed. Graphs were created in GraphPad Prism (9.3.1) and exported as an enhanced metafile for integration into figures that were compiled in Adobe Illustrator (24.3). All data are represented as the mean ± the standard error of the mean. Illustrations were created in their entirety in Adobe Illustrator.

## Code/software

Custom Excel modules (created in Excel's Visual Basic Editor) were used for the analysis of cytoplasmic and mitochondrial calcium imaging. The modules are available online at https://github.com/rachel-doser/GCaMP_Analysis_Excel_VBA (*Doser, 2021*).

## Acknowledgements

We thank Sasha de Henau for the mito-roGFP plasmid, Attila Stetak for the GCaMP6f plasmid, and the Caenorhabditis Genetics Center at the University of Minnesota for strains. This work was largely supported in part by an R01 awarded to FJ Hoerndli from NIH/NINDS (NS115947). Additional financial support was provided by the College of Veterinary Medicine and Biomedical Sciences at Colorado State University. *C. elegans* strains were purchased from the Caenorhabditis Genetics Center, which is funded by the NIH Office of Research Infrastructure Programs (P40 OD010440).

## Additional information

### Funding

| Funder | Grant reference number | Author |
|---|---|---|
| National Institute of Neurological Disorders and Stroke | NS115947 | Frederic J Hoerndli |
| CVMBS, Colorado State | College Research Council grant | Rachel L Doser |

The funders had no role in study design, data collection and interpretation, or the decision to submit the work for publication.

### Author contributions

Rachel L Doser, Conceptualization, Data curation, Formal analysis, Validation, Investigation, Methodology, Writing - original draft, Writing - review and editing; Kaz M Knight, Conceptualization, Data

curation, Formal analysis, Validation, Investigation, Methodology, Writing - review and editing; Ennis W Deihl, Data curation, Validation, Investigation, Methodology, Writing - review and editing; Frederic J Hoerndli, Conceptualization, Data curation, Funding acquisition, Investigation, Methodology, Writing - original draft, Project administration, Writing - review and editing

**Author ORCIDs**
Rachel L Doser ⓘ http://orcid.org/0000-0001-9057-4371
Kaz M Knight ⓘ http://orcid.org/0000-0002-3184-3620
Ennis W Deihl ⓘ http://orcid.org/0009-0003-6737-8964
Frederic J Hoerndli ⓘ http://orcid.org/0000-0001-6838-0386

**Decision letter and Author response**
Decision letter https://doi.org/10.7554/eLife.92376.sa1
Author response https://doi.org/10.7554/eLife.92376.sa2

---

## Additional files

**Supplementary files**
• MDAR checklist

### Data availability

All data generated or analysed during this study are accessible on Dryad under the following DOI https://doi.org/10.5061/dryad.0gb5mkm71.

The following dataset was generated:

| Author(s) | Year | Dataset title | Dataset URL | Database and Identifier |
|---|---|---|---|---|
| Doser R, Knight K, Deihl E, Hoerndli FJ | 2024 | Image quantification data for: Activity-dependent mitochondrial ROS signaling regulates recruitment of glutamate receptors to synapses | https://dx.doi.org/10.5061/dryad.0gb5mkm71 | Dryad Digital Repository, 10.5061/dryad.0gb5mkm71 |

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

**Appendix 1**

## Appendix 1—key resources table

| Reagent type (species) or resource | Designation | Source or reference | Identifiers | Additional information |
|---|---|---|---|---|
| Strain, strain background (*Caenorhabditis elegans*) | AVA GLR-1::GFP (transgenic strain) | Hoerndli Lab, CSU | FJH 18 | Genotype: *akIs141* II; *glr-1(ky176)* III *akIs141* contents: *Prig-3::GLR-1::GFP* (integrated) |
| Strain, strain background (*C. elegans*) | AVA SEP::GLR-1 (transgenic strain) | Hoerndli Lab, CSU | FJH 314 | Genotype: *akIs172; glr-1(ky176)* III *akIs172* contents: *Prig-3::SEP::GLR-1 + Peat-4::ChR2::mCherry (integrated)* |
| Strain, strain background (*C. elegans*) | AVA ChRimson and mito-roGFP (transgenic strain) | This paper | FJH 402 | Genotype: *lin-15(n765ts)* X; *lite-1(ok530)* X; *csfEx160 csfEx160* contents: pRD30 + pRD15 + pJM23 + pCT61 |
| Strain, strain background (*C. elegans*) | AVA ChRimson and cyto-GCaMP (transgenic strain) | This paper | FJH 412 | Genotype: *lin-15(n765ts)* X; *lite-1(ok530)* X; *csfEx167 csfEx167* contents: pRD30 + pAS1 + pJM23 |
| Strain, strain background (*C. elegans*) | AVA mitoKR and mito-roGFP (transgenic strain) | This paper | FJH 416 | Genotype: *lin-15(n765ts)* X; *lite-1(ok530)* X; *csfEx168 csfEx168* contents: pRD36 + pRD15 + pJM23 |
| Strain, strain background (*C. elegans*) | AVA GLR-1::GFP and mitoKR (transgenic strain) | This paper | FJH 555 | Genotype: *akIs141* II; *glr-1(ky176)* III; *csfEx188 csfEx188* contents: pRD36 + pJM23 + pCT61 |
| Strain, strain background (*C. elegans*) | AVA GLR-1::GFP in *mcu-1(lf)* (transgenic strain) | This paper | FJH 576 | Genotype: *akIs141* II; *glr-1(ky176)* III; *mcu-1(ju1154)* IV |
| Strain, strain background (*C. elegans*) | AVA SEP::GLR-1 with mitoKR (transgenic strain) | This paper | FJH 582 | Genotype: *lin-15(n765ts)* X; *glr-1(ky176)* III *csfEx210 csfEx210* contents: pRD36 + pDM1442 + pJM23 |
| Strain, strain background (*C. elegans*) | AVA SEP::GLR-1 (transgenic strain) | This paper | FJH 635 | Genotype: *lin-15(n765ts)* X; *glr-1(ky176)* III *csfEx234 csfEx234* contents: pDM1442 + pJM23 + pCT61 |
| Strain, strain background (*C. elegans*) | AVA SEP::GLR-1 in *mcu-1(lf)* (transgenic strain) | This paper | FJH 638 | Genotype: *akIs172* II; *glr-1(ky176)* III; *mcu-1(ju1154)* IV |
| Strain, strain background (*C. elegans*) | AVA ChRimson and GCaMP in *mcu-1(lf)* (transgenic strain) | This paper | FJH 641 | Genotype: *lin-15(n765ts)* X; *mcu-1(ju1154)* IV; *csfEx261 csfEx261* contents: pRD30 + pAS1 + pJM23 + pCT61 |
| Strain, strain background (*C. elegans*) | AVA ChRimson and mito-GCaMP (transgenic strain) | This paper | FJH 644 | Genotype: *lin-15(n765ts), lite-1(ok530)* X; *csfEx264 csfEx264* contents: pKK01 + pRD30 + pJM23 + pCT61 |
| Strain, strain background (*C. elegans*) | AVA ChRimson and mito-GCaMP in *mcu-1(lf)* (transgenic strain) | This paper | FJH 647 | Genotype: *lin-15(n765ts), lite-1(ok530)* X; *mcu-1(ju1154)* IV; *csfEx264 csfEx264* contents: see above |
| Strain, strain background (*C. elegans*) | AVA SEP::GLR-1 and mito-TdTom. (transgenic strain) | This paper | FJH 690 | Genotype: *glr-1(ky176)* III; *csfEx268 csfEx268* contents: pKK07 + pDM1442 + pCT61 |
| Strain, strain background (*C. elegans*) | AVA ChRimson and mito-roGFP (transgenic strain) | This paper | FJH 706 | Genotype: *lite-1(ok530)* X; *mcu-1(ju1154)* IV; *csfEx160 csfEx160* contents: pRD30 + pRD15 + pJM23 + pCT61 |
| Recombinant DNA reagent | AVA mito-roGFP (plasmid) | This paper | pRD15 | *Pflp-18::TOMM-20::roGFP::let-858 5'UTR* |
| Recombinant DNA reagent | AVA ChRimson TdTomato (plasmid) | This paper | pRD30 | *Pflp-18::ChRimson::tdTomato::let-858 5'UTR* |
| Recombinant DNA reagent | AVA mitoKR (plasmid) | This paper | pRD36 | *Pflp-18::TOMM-20::KillerRed::let-858 5'UTR* |
| Recombinant DNA reagent | AVA mito-GCaMP (plasmid) | This paper | pKK01 | *Pflp-18::mito4x-GCaMP6f::let-858 5'UTR* |
| Recombinant DNA reagent | AVA mito-TdTomato (plasmid) | This paper | pKK07 | *Pflp-18::TOMM-20::tdTomato::let-858 5'UTR* |
| Recombinant DNA reagent | AVA ChRimson mCherry (plasmid) | This paper | pED01 | *Pflp18::ChRimson::mCherry* |
| Recombinant DNA reagent | AVA cytoplasmic GCaMP (plasmid) | Stetak Lab, University of Zurich | pAS1 | *Prig-3::GCaMP6f::unc-54 5'UTR* |
| Recombinant DNA reagent | AVA SEP-tagged GLR-1 (plasmid) | Maricq Lab, University of Utah | pDM1442 | *Prig-3::SEP::GLR-1::unc-54 5'UTR* |
| Recombinant DNA reagent | Lin-15 rescue (plasmid) | Maricq Lab, University of Utah | pJM23 | *Plin-15::lin-15+* |

*Appendix 1—key resources table continued on next page*

## Appendix 1—key resources table continued

| Reagent type (species) or resource | Designation | Source or reference | Identifiers | Additional information |
|---|---|---|---|---|
| Recombinant DNA reagent | Filler DNA (plasmid) | Stratagene | pBSKS | For plasmid recombination into extrachromosomal array |
| Recombinant DNA reagent | Co-injection marker (plasmid) | Hoerndli Lab, CSU | pCT61 | Pegl-20::nls::DsRed |
| Recombinant DNA reagent | AVA mCherry tagged SOL-1 (plasmid) | Maricq Lab, University of Utah | pWR38 | Prig-3::sol-2::mCherry |
| Recombinant DNA reagent | AVA Gateway vector (plasmid) | Hoerndli Lab, CSU | pCT22 | Gateway pENTR [4-1] Pflp-18 promoter |
| Recombinant DNA reagent | ChRimson Gateway vector (plasmid) | Hoerndli Lab, CSU | pFH13 | Gateway pENTR12 ChRimson no STOP |
| Recombinant DNA reagent | TdTomato Gateway vector (plasmid) | Hoerndli Lab, CSU | pGH162 | Gateway [2-3] tdTomato_let858UTR |
| Recombinant DNA reagent | Destination Gateway vector (plasmid) | Jorgensen Lab, University of Utah | pCFJ150 | pDEST/expression vector |
| Recombinant DNA reagent | 3′ UTR Gateway vector (plasmid) | Hoerndli Lab, CSU | pFH21 | Gateway pENTR [2-3] 3′UTR (let-858) |
| Recombinant DNA reagent | Mito-roGFP Gateway vector (plasmid) | This paper | pRD02 | Gateway pENTR [2-1] TOMM-20 roGFP |
| Recombinant DNA reagent | AVA cytoplasmic KillerRed (plasmid) | This paper | pRD22 | Pflp-18::KillerRed::let858 |
| Recombinant DNA reagent | Mito-GCaMP Source (plasmid) | Addgene | Addgene plasmid no. 127870 | CMV:Mito4x-GCaMP6f |
| Recombinant DNA reagent | Mito-roGFP Source (plasmid) | De Henau Lab, University Medical Center | pSHD1 | Pflbf1::TOMM20::roGFP2Tsa2::3′UTRtbb2 |
| Sequence-based reagent | PCR primers for cloning pRD15 | This paper | pSDH1_F | ggggacaagtttgtacaaaaaagcaggctGACatgagctccaccgtg |
| Sequence-based reagent | PCR primers for cloning pRD15 | This paper | pSDH1_R | ggggaccactttgtacaagaaagctgggtgcttgaaaggatcttgcattt |
| Sequence-based reagent | PCR primers for cloning pRD36 | This paper | pRD15_F | CTTGTACAAAGTGGTTGGATGATCG |
| Sequence-based reagent | PCR primers for cloning pRD36 | This paper | pRD15_R | TGCTCCAGCCTGGGCACG |
| Sequence-based reagent | PCR primers for cloning pRD36 | This paper | pRD22_F | GCCCAGGCTGGAGCATCCGAGGG AGGCCCAGCC |
| Sequence-based reagent | PCR primers for cloning pRD36 | This paper | pRD22_R | ACCACTTTGTACAAGTAATCCTCGTCGGATCCGATGG |
| Sequence-based reagent | PCR primers for cloning pKK01 | This paper | pRD15_F2 | CTTGTACAAAGTGGTTGGATGA |
| Sequence-based reagent | PCR primers for cloning pKK01 | This paper | pRD15_R2 | GTCATGTCTAACCCTGAAATT |
| Sequence-based reagent | PCR primers for cloning pKK01 | This paper | AG127870_F | AGGGTTAGACATGACATGAGCGTGCTGACACCTCTG |
| Sequence-based reagent | PCR primers for cloning pKK07 | This paper | AG127870_R | ACCACTTTGTACAAGCTGATCAGCGGGTTAAACGGG |
| Sequence-based reagent | PCR primers for cloning pKK07 | This paper | pRD15_F3 | CTTGTACAAAGTGGTTGGATGATCG |
| Sequence-based reagent | PCR primers for cloning pKK07 | This paper | pRD15_R3 | TGCTCCAGCCTGGGCACG |
| Sequence-based reagent | PCR primers for cloning pKK07 | This paper | pGH162_F | GCCCAGGCTGGAGCATGGTGAGC AAGGGGCGAGG |
| Sequence-based reagent | PCR primers for cloning pKK07 | This paper | pGH162_R | ACCACTTTGTACAAGTTACTTGTACAGCTCGTCCATGCC |
| Sequence-based reagent | PCR primers for cloning pED01 | This paper | pRD30_F | GGATGATCGACGCCAACGT |
| Sequence-based reagent | PCR primers for cloning pED01 | This paper | pRD30_R | CCACTTTGTACAAGAAAGCTGGGT |
| Sequence-based reagent | PCR primers for cloning pED01 | This paper | pWR80_F | TCTTGTACAAAGTGGTGGTCTCAAAGGGTGAAGAAG |
| Sequence-based reagent | PCR primers for cloning pED01 | This paper | pWR80_R | TGGCGTCGATCATCCCACCATATTCCTTATACAATTCATC |

## Appendix 2

## FRAP assays of tagged GLR-1

*Appendix 2—figure 1A* illustrates the differential effect of photobleaching (PB) on SEP-tagged GLR-1 based on subcellular location. Recovery of SEP fluorescence after PB is indicative of the balance between GLR-1 recruitment (i.e., via exocytosis from transport vesicles or synaptic endosomes) and recycling (i.e., via receptor endocytosis). As illustrated in *Appendix 2—figure 1B*, GLR-1::GFP allows for visualization of all GLR-1 molecules, including those positioned at the synaptic membrane or in endosomes. Following PB of GFP, the fluorescence recovery indicates that new GLR-1 has been transported and delivered to the synaptic membrane or endosome within the region of interest.

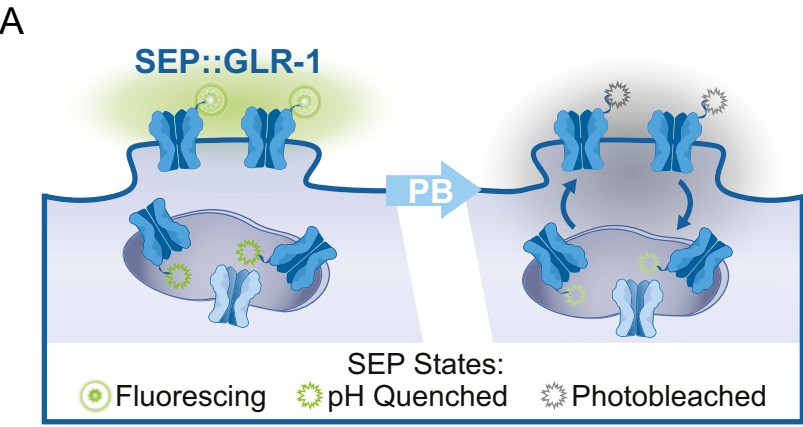

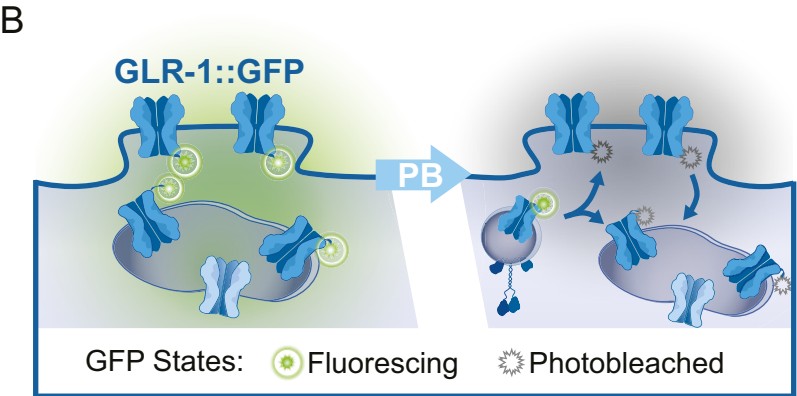

**Appendix 2—figure 1.** FRAP assays of tagged GLR-1. (**A**) Illustration of subcellular SEP::GLR-1 localization. Following photobleaching (PB) of fluorescing SEP (attached to GLR-1 positioned at the synaptic membrane), recovery of SEP fluorescence is indicative of the rate of GLR-1 exocytosis from transport vesicles or synaptic endosomes and receptor endocytosis. (**B**) Illustration depicting the localization of GLR-1::GFP to the synaptic membrane or in endosomes. Following PB of GFP, the fluorescence recovery indicates that new GLR-1 has been transported and delivered to the synaptic membrane or endosome within the region of interest.

