## [Editor Report]

This study examines an interplay between synaptic mitochondria and glutamate receptor exocytosis in *C. elegans*. Collectively, the solid results support the idea that mitochondrial function influences receptor dynamics at postsynaptic sites. This is important because tight control of synaptic function likely integrates several mitochondrial functions: energy production, calcium buffering, and (here) reactive oxygen species signaling.

---

## [Decision Letter]

**Decision letter after peer review:**

Thank you for submitting your article "Activity-dependent Mitochondrial ROS Signaling Regulates Recruitment of Glutamate Receptors to Synapses" for consideration by *eLife*. Your article has been reviewed by 3 peer reviewers, and the evaluation has been overseen by a Reviewing Editor and Lu Chen as the Senior Editor.

Essential revisions:

The following key revisions are expected in a revised manuscript:

1) There appears to be a mismatch between the text and the data presented in Figure 2.

2) The Framing of the study: There does not seem to be a coherent mechanistic connection from calcium to MCU to mitochondria to ROS, though the text in the manuscript implies this. The authors should revise both the results and the Discussion sections to clarify their findings.

3) Several key experiments are suggested to strengthen and expand this study. This includes depleting ROS in the system to confirm the proposed model. Also, given that the KillerRed generation of ROS resulted in a reduction in all parameters measured, there is a concern that a large amount of ROS generated might lead to toxicity. Perhaps using another method to manipulate ROS would be helpful. Examining another cellular process that is not altered would help validate that the KillerRed is not toxic, such as looking at the transport of at least one other protein.

*Reviewer #2 (Recommendations for the authors):*

Text revisions would be sufficient for almost all of the issues and questions delineated above. Figure 2 should be clarified, so the data match the text.

Optionally, the authors could round out experiments (more mechanical stimulation, other modes of enhancing ROS, scavenging ROS to see if there is a phenotype reversal, etc.). Those kinds of tests could potentially bolster the story (though they would not change the core conclusions).

*Reviewer #3 (Recommendations for the authors):*

1. The authors found that optical activation of Chrimson expressed in AVA leads to increased calcium uptake into mitochondria based on mitoGCaMP signal and this increase is dependent on the mitochondrial calcium transporter mcu-1 (Figure 1). The authors found that GCaMP signal increases to varying amounts in different regions of the dendrite and after each stimulus suggesting that mitochondria are diverse. The differences in GCaMP signal for the different clusters of mitochondria could be due to changes in the numbers of mitochondria in each cluster (if they redistribute after each stimulus) or perhaps due to the proximity of the mitochondria to calcium channels. The authors should consider analyzing the distribution of mitochondria in AVA after repeated stimuli and the relationship of the mitochondria to VGCCs? One might expect mitochondrial clusters closest to VGCCs to have higher increases in mitoGCaMP signal after stimulation.

2. The authors state that (line 86) "most dendritic mitochondria were located in close proximity to clusters of surface localized GLR-1…." If ROS have a range of action of 1um, it would be informative to know what percentage of mitochondria are within 1um of GLR-1-GFP.

3. In Figure 2, the authors show that blocking calcium uptake into mitochondria leads to increased exocytosis of GLR-1 (as measured by FRAP of SEP-GLR-1) suggesting that increased calcium in mitochondria is important for regulating GLR-1. However, it remains possible that the inability of mitochondria to take up calcium results in a local increase in cytoplasmic calcium, which then promotes exocytosis of GLR-1. Although the authors show that cytoplasmic GCaMP6f fluorescence does not change under these conditions, GCaMP6f may not be sensitive enough to report local changes in cytoplasmic calcium. Also, although the peak GCaMP6f signal under the various conditions in Figure S2F are similar, the shapes of the curves appear to differ with mcu-1(lf) and Ru360 having prolonged tails indicative of higher residual cytoplasmic calcium. Is it possible to test whether sequestering cytoplasmic calcium near mitochondria affects FRAP of SEP-GLR-1 in mcu-1 genetic mutants?

4. For Figure 2D and 2E, the authors conclude that (line 165): "… the rate of GLR-1-GFP FRAP in mcu-1(lf) was comparable to controls whereas the rate of FRAP in Ru360 treated animals was slightly increased." However, Figure 2E shows a significant decrease in FRAP of GLR-1-GFP in mcu-1 genetic mutants.

5. The GLR-1-GFP clusters shown in the "Before" images in mcu-1(lf) appear larger and more defined than in the Control image (Figure 2D). Does mcu-1(lf) affect synapse size or number?

6. Given that calcium uptake into mitochondria can stimulate ATP production, is it possible that less ATP production in mcu-1(lf) results in decreased GLR-1 transport events? Can the authors measure the transport of another unrelated cargo in mcu-1(lf) to test if motor-dependent transport is generally altered?

7. The authors should add to the discussion a few sentences about known mechanisms of how calcium uptake into mitochondria leads to ROS production.

8. Figure 4 shows that photoactivation of mitoKR leads to a reduction in all parameters tested (decreased FRAP of SEP-GLR-1, decreased FRAP of GLR-1-GFP and decreased FRAP of GLR-1 transport) raising concern that the levels of ROS generated by mitoKR may be non-specifically toxic to cellular processes. Are the levels of ROS generated after photoactivation of mitoKR comparable to the ROS levels observed after neuronal stimulation (as shown in Figure 3)?

9. The GLR-1-GFP clusters in the "Before" images in Figure 4D look altered in mitoKR versus control. Is GLR-1-GFP intensity and distribution altered in *C. elegans* expressing mitoKR?

10. The mcu-1(lf) genetic mutant has a stronger effect on FRAP of SEP-GLR-1 than the drug Ru360. The experiment shown in Figure 5C would be more compelling if the mitoKR activation was performed in mcu-1(lf).

11. The model that increased uptake of calcium into mitochondria via mcu-1 leads to increased mitochondrial production of ROS, and that the increase in ROS is responsible for regulating GLR-1 would be stronger if a loss of function ROS experiment could be performed. The concern is that blocking calcium entry into mitochondria in mcu-1(lf) could affect other mitochondrial functions in addition to loss of ROS that could be responsible for the effects of mcu-1(lf) on GLR-1 trafficking. If ChRimson activation leads to decreased FRAP of SEP-GLR-1, then one could test if depleting ROS with a scavenger or preventing mitochondrial ROS generation blocked the effects of neuronal activation on GLR-1 exocytosis. Alternatively, does depletion of ROS lead to increased FRAP of SEP as observed in mcu-1(lf) genetic mutants (Figure 2)? If the model is correct, depletion of ROS would be predicted to mimic the effects of mcu-1(lf) on GLR-1 trafficking.

*Reviewer #4 (Recommendations for the authors):*

The term "FRAP rate" is not clearly defined in the manuscript, leaving the exact comparisons ambiguous. Evaluating metrics like time constants, diffusion coefficients, and mobile fractions derived from the FRAP recovery curve could offer deeper insights into the endosomal trafficking of GLR-GFP.

The use of GLR-SEP is not validated within the manuscript. It would be beneficial if the authors included videos of the FRAP experiments utilizing GLR-SEP. Given that GLR-SEP solely labels AMPA receptors on the cell surface, these videos should primarily display exocytosis events, excluding endosomal trafficking. Quantifying the number or frequency of these exocytosis events would provide a more direct assessment of AMPA receptor exocytosis. The "FRAP" recovery encompasses exocytosis, endocytosis, and lateral diffusion, which might not accurately capture alterations in exocytosis alone.

The authors observed an increase in raw fluorescence recovery for GLR-SEP in the MCU-1 mutant compared to the wild-type (Figure 2C), yet found that the percentage of recovery remained consistent in the mutant condition (Figure S2C). They inferred an elevation in exocytosis in the MCU-1 mutant based on these findings. However, there are concerns with this conclusion. Firstly, the FRAP experiments do not offer a direct assessment of exocytosis. Secondly, if the total expression level of GLR-SEP is augmented in the MCU-1 mutant, the raw fluorescence recovery could inherently be faster than that of the wild-type. The inclusion of appropriate controls is essential to validate these outcomes more robustly.

Figure 5B shows an enhanced % FRAP recovery for GLR-SEP in the control+PA Ru360 compared to the untreated group. However, Figure S2C doesn't indicate any difference between the Ru360 and untreated groups. Could the authors elucidate the inconsistency observed between these datasets?

There are discrepancies observed in the data presented. In Figure 2E and 2G, both FRAP recovery and trafficking events of GLR-GFP appear reduced in the MCU-1 mutant. Conversely, Figure 4E and 4H show a reduction in FRAP recovery and trafficking in the mitoKR condition. Given that ROS is diminished in the MCU-1 mutant but elevated during mitoKR global activation, can the authors elucidate and discuss these findings in greater detail? The current working model lacks clarity.

A clearer presentation might involve splitting the working model in Figure 6 into two distinct parts. One should depict the effect of ROS on AMPA receptor dendritic trafficking, while the other illustrates its influence on AMPA receptor surface expression/excocytosis. Given the potential differential impacts of mitochondrial calcium and ROS on these mechanisms, delineating the models for each process separately would enhance comprehension of the findings.

---

## [Author Response]

Essential revisions:The following key revisions are expected in a revised manuscript:1) There appears to be a mismatch between the text and the data presented in Figure 2.

We thank the reviewers and editors for their detailed review of our manuscript. This unintentional mismatch has been fixed with changes to the manuscript text.

2) The Framing of the study: There does not seem to be a coherent mechanistic connection from calcium to MCU to mitochondria to ROS, though the text in the manuscript implies this. The authors should revise both the results and the Discussion sections to clarify their findings.

We agree that we were not able to elucidate all relevant players involved in the proposed mechanism. So, it is indeed more appropriate to explicitly state what parts of the proposed mechanism are supported by the data presented and what mechanistic steps are speculative based on what is known in the field of mitochondrial calcium (Ca^2+^) handling and redox signaling. We have made many changes to the manuscript text and the illustration of our proposed model to be more careful in making this distinction and to not overstate the implications of our findings.

3) Several key experiments are suggested to strengthen and expand this study. This includes depleting ROS in the system to confirm the proposed model. Also, given that the KillerRed generation of ROS resulted in a reduction in all parameters measured, there is a concern that a large amount of ROS generated might lead to toxicity. Perhaps using another method to manipulate ROS would be helpful. Examining another cellular process that is not altered would help validate that the KillerRed is not toxic, such as looking at the transport of at least one other protein.

These are all important experiments and many of which we have already conducted in one way or another. First, we have published a short report on the effects of ROS scavenging on GLR-1 transport (PMID: 35622512). In brief, we found that like the effect of increased ROS levels, genetic and strong pharmacological ROS scavenging also caused a decrease in GLR-1 transport. One of the major caveats with our previous experiments, and others that would address the point made by the reviewers, is the lack of sensitivity of current in vivo ROS sensors such as roGFP and HyPer, which do not allow establishing “low ROS” conditions within physiological range. In addition to limited sensitivity of in vivo ROS sensors, current genetic and pharmacological ROS scavenging approaches, lack cell specificity or temporal control needed for these experiments.

Second, the concern about KillerRed toxicity is valid especially since it was designed to generate relatively high levels of ROS. We worked for a long time to determine appropriate expression levels and activation protocols for KillerRed. The optical activation protocols for KillerRed used in this study were optimized to results in ROS elevations that are similar to what is elicited by native neuronal activity (e.g., resulting from mechanosensation). The side-by-side comparison of mito-roGFP measurements resulting from mechanosensory stimulation, and local or global mitoKillerRed activation demonstrates that KillerRed activation causes ROS elevations that are comparable to those that arise normally following repetitive neuronal activation. We have rearranged the main text, figures, and figure supplements to allow for these important validations to be discussed and shown as a main figure (Figure 4).

Lastly, we agree that assessing if mitoROS alters the transport of other proteins is important and would not only further support our KillerRed activation protocol, but also provide insight into the specificity of mitoROS signaling on GLR-1 transport. However, this requires the creation and validation of *C. elegans* strains with cell-specific expression of fluorescently labeled proteins (e.g., NMDA receptors such as NMR-1/2) that we currently do not have. We plan to develop these strains and carry out follow-up experiments with KillerRed and mitoROS that would test our reagents, conditions, and model. These would be published as follow-up or stand-alone studies depending on their content and significance. In the meantime, however, we have analyzed transport velocities of GLR-1, which to some extent reflect availability of ATP and cytoskeletal integrity in *mcu-1* mutants and mitoKR conditions (revised Figure 2- supplement 1C, Figure 5 supplement 1 A-B). These analyses show that transport velocities are unchanged in any of the conditions above suggesting no toxic effects of these conditions on kinesin-mediated transport.

Reviewer #2 (Recommendations for the authors):Text revisions would be sufficient for almost all of the issues and questions delineated above. Figure 2 should be clarified, so the data match the text.Optionally, the authors could round out experiments (more mechanical stimulation, other modes of enhancing ROS, scavenging ROS to see if there is a phenotype reversal, etc.). Those kinds of tests could potentially bolster the story (though they would not change the core conclusions).

We thank the reviewer for suggesting text changes to clarify our results and we hope we have addressed their concerns (see detailed responses above).

We agree that using mechanical stimulation or other methods for increasing mitoROS to further support this mechanism would be ideal. The current available reagent and approaches, lack sensitivity, cell specificity, subcellular localization control or temporal control or a combination of these. We are currently developing new reagents and approaches that will test our model and follow up on the results we present here.

Reviewer #3 (Recommendations for the authors):1. The authors found that optical activation of Chrimson expressed in AVA leads to increased calcium uptake into mitochondria based on mitoGCaMP signal and this increase is dependent on the mitochondrial calcium transporter mcu-1 (Figure 1). The authors found that GCaMP signal increases to varying amounts in different regions of the dendrite and after each stimulus suggesting that mitochondria are diverse. The differences in GCaMP signal for the different clusters of mitochondria could be due to changes in the numbers of mitochondria in each cluster (if they redistribute after each stimulus) or perhaps due to the proximity of the mitochondria to calcium channels. The authors should consider analyzing the distribution of mitochondria in AVA after repeated stimuli and the relationship of the mitochondria to VGCCs? One might expect mitochondrial clusters closest to VGCCs to have higher increases in mitoGCaMP signal after stimulation.

We agree that investigating mitochondrial positioning and how it would shape mitochondrial function and calcium propagation would be interesting and insightful work. However, this warrants a follow-up study of its own. Indeed, we are currently in the process of deepening our analysis of mitochondrial distribution, morphology and Ca^2+^ propagation in vivo using some of the tools developed for this study.

2. The authors state that (line 86) "most dendritic mitochondria were located in close proximity to clusters of surface localized GLR-1…." If ROS have a range of action of 1um, it would be informative to know what percentage of mitochondria are within 1um of GLR-1-GFP.

We have done this analysis and included in the results (at line 163-164) that around 61% of mitochondria are located within 1 µm of a SEP::GLR-1 cluster/puncta. However, this is only an estimation because the two AVA neurites run directly adjacent to one another, so we are unable to precisely distinguish whether GLR-1 clusters and mitochondria reside within the same AVA neurite. Additionally, this estimation should not be construed to mean that about 61% mitochondria are located near glutamatergic synapses because GLR-1 clusters exist in these neurites without a presynaptic terminal nearby (PMID: 12123612).

3. In Figure 2, the authors show that blocking calcium uptake into mitochondria leads to increased exocytosis of GLR-1 (as measured by FRAP of SEP-GLR-1) suggesting that increased calcium in mitochondria is important for regulating GLR-1. However, it remains possible that the inability of mitochondria to take up calcium results in a local increase in cytoplasmic calcium, which then promotes exocytosis of GLR-1. Although the authors show that cytoplasmic GCaMP6f fluorescence does not change under these conditions, GCaMP6f may not be sensitive enough to report local changes in cytoplasmic calcium. Also, although the peak GCaMP6f signal under the various conditions in Figure S2F are similar, the shapes of the curves appear to differ with mcu-1(lf) and Ru360 having prolonged tails indicative of higher residual cytoplasmic calcium. Is it possible to test whether sequestering cytoplasmic calcium near mitochondria affects FRAP of SEP-GLR-1 in mcu-1 genetic mutants?

The sensitivity of GCaMP6f could surely occlude subtle changes in cytoplasmic calcium when *mcu-1* is absent or blocked. We also noticed that some GCaMP peaks had a prolonged shoulder suggesting delayed calcium buffering, but this feature was not consistently observed in our dataset and was not robust enough to be detected by our ‘Total Activity’ measurement (revised Figure 2I-J) which would detect changes in the duration of ca^2+^ events. We could employ a more sensitive calcium indicator (i.e., jGCaMP8), and are currently developing transgenic strains and imaging approaches to follow up on our result in this study. We are unaware of current genetically encoded reagents that would help sequester calcium locally at mitochondria with subcellular and temporal control.

4. For Figure 2D and 2E, the authors conclude that (line 165): "… the rate of GLR-1-GFP FRAP in mcu-1(lf) was comparable to controls whereas the rate of FRAP in Ru360 treated animals was slightly increased." However, Figure 2E shows a significant decrease in FRAP of GLR-1-GFP in mcu-1 genetic mutants.

The confusion is due to an incorrect statement in the results text. We have corrected this error and appreciate the reviewer for bringing it to our attention.

5. The GLR-1-GFP clusters shown in the "Before" images in mcu-1(lf) appear larger and more defined than in the Control image (Figure 2D). Does mcu-1(lf) affect synapse size or number?

We observed a slight but not significant increase in the size and intensity GLR-1::GFP puncta in *mcu-(lf)* mutants (data not shown), but puncta number remained comparable to controls.

6. Given that calcium uptake into mitochondria can stimulate ATP production, is it possible that less ATP production in mcu-1(lf) results in decreased GLR-1 transport events? Can the authors measure the transport of another unrelated cargo in mcu-1(lf) to test if motor-dependent transport is generally altered?

This is possible and although we could do an additional experiment to analyze transport of another unrelated cargo, we have added supplemental transport velocity data (revised Figure 2 -supplement 1C) that suggests ATP levels are comparable between *mcu-1(lf)* and controls. The processivity of molecular motors requires a consistent supply of ATP. Thus, if ATP was decreased in *mcu-1(lf),* then transport would occur at a slower rate. Instead, we found that transport velocities were nearly identical between *mcu-1(lf)* and controls.

7. The authors should add to the discussion a few sentences about known mechanisms of how calcium uptake into mitochondria leads to ROS production.

We have added discussion and some additional citations to reviews on this topic.

8. Figure 4 shows that photoactivation of mitoKR leads to a reduction in all parameters tested (decreased FRAP of SEP-GLR-1, decreased FRAP of GLR-1-GFP and decreased FRAP of GLR-1 transport) raising concern that the levels of ROS generated by mitoKR may be non-specifically toxic to cellular processes. Are the levels of ROS generated after photoactivation of mitoKR comparable to the ROS levels observed after neuronal stimulation (as shown in Figure 3)?

Yes, our KillerRed photoactivation protocol increases ROS at mitochondria to a similar level as neuronal activation. More specifically, 10 minutes of mechano-stimulation (revised Figure 4A-B) and 5 minutes of repetitive optical stimulation (revised Figure 3B-C; see more detail in response to Reviewer 2 – Recommendations for the authors) led to an approximate doubling of the roGFP F_ratio_ as did both of our local photoactivation protocol for mitoKR activation (Figure 4C-E). Our 10-minute global photoactivation of mitoKR caused only a modest increase in the roGFP F_ratio_ (~30% increase) (revised Figure 4G-H). This data is now shown in a new Figure 4 to explicitly show that our mitoKR activation protocol increases mitoROS to a similar or even lesser extent than what is elicited by neuronal activity.

9. The GLR-1-GFP clusters in the "Before" images in Figure 4D look altered in mitoKR versus control. Is GLR-1-GFP intensity and distribution altered in *C. elegans* expressing mitoKR?

We did not detect any changes in the intensity or density of GLR-1::GFP puncta with mitoKR (data not shown) or increased genetically (PMID: 32847966). This difference is probably due to the natural heterogeneity we see in GLR-1 cluster localization and density which can be also seen in Figure 2 and in our previously published work (PMIDs: 32847966, 25843407).

10. The mcu-1(lf) genetic mutant has a stronger effect on FRAP of SEP-GLR-1 than the drug Ru360. The experiment shown in Figure 5C would be more compelling if the mitoKR activation was performed in mcu-1(lf).

We understand why the reviewer would suggest combining *mcu-1(lf)* and mitoKR based on the severity of the *mcu-1(lf)* phenotype. However, we reasoned it would make more sense to combine acute treatments (Ru360 and mitoKR activation) without possible developmental changes generally associated with genetic *loss-of-functions*. More specifically, *mcu-1(lf)* led to increased synaptic GLR-1 puncta (revised Figure2-supplement 1A) that was more severe than acute Ru360 treatment, so we worried that these changes were at least partially developmental and unlikely to be compensated by an acute photoactivation of mitoKR.

11. The model that increased uptake of calcium into mitochondria via mcu-1 leads to increased mitochondrial production of ROS, and that the increase in ROS is responsible for regulating GLR-1 would be stronger if a loss of function ROS experiment could be performed. The concern is that blocking calcium entry into mitochondria in mcu-1(lf) could affect other mitochondrial functions in addition to loss of ROS that could be responsible for the effects of mcu-1(lf) on GLR-1 trafficking. If ChRimson activation leads to decreased FRAP of SEP-GLR-1, then one could test if depleting ROS with a scavenger or preventing mitochondrial ROS generation blocked the effects of neuronal activation on GLR-1 exocytosis. Alternatively, does depletion of ROS lead to increased FRAP of SEP as observed in mcu-1(lf) genetic mutants (Figure 2)? If the model is correct, depletion of ROS would be predicted to mimic the effects of mcu-1(lf) on GLR-1 trafficking.

One of the main other “mitochondrial functions” that could be impacted by loss of *mcu-1(lf)* would be ATP production. We have added GLR-1 transport velocity data showing no change in transport velocity indicating that ATP levels are comparable between *mcu-1(lf)* and controls (revised Figure 2-supplement 1C). We agree that the concern of ROS-independent effects in *mcu-1(lf)* would be reduced by data showing decreased ROS levels increase synaptic delivery and exocytosis in a similar fashion to *mcu-1(lf).* We have not done these experiments but is something we could do. However, we would not be able to validate our methods for reducing ROS because the ROS sensitivity range of mito-roGFP is too high to detect decreases in basal mitoROS (meaning below 10 nM). We have shown that genetic and pharmacological means of decreasing ROS decreased GLR-1 transport (opposite from what was expected) but via a mechanism independent of cytoplasmic ca^2+^ signaling (PMID: 35622512). Addressing this will depend on the development of more sensitive ROS indicators necessary to test the efficacy of antioxidant treatments in vivo.

Reviewer #4 (Recommendations for the authors):The term "FRAP rate" is not clearly defined in the manuscript, leaving the exact comparisons ambiguous. Evaluating metrics like time constants, diffusion coefficients, and mobile fractions derived from the FRAP recovery curve could offer deeper insights into the endosomal trafficking of GLR-GFP.

We thank the reviewer for this insightful comment. We do agree that the term “FRAP rate” should be described in better detail. We did not obtain accurate measures of time constants, mobile fractions, and diffusion coefficients because the time required to reach a steady-state FRAP for GLR-1 (i.e., a plateau in the % FRAP recovery curve) requires long imaging durations (> 45 min.) that are problematic in vivo due to hypoxia and starvation.

The use of GLR-SEP is not validated within the manuscript. It would be beneficial if the authors included videos of the FRAP experiments utilizing GLR-SEP. Given that GLR-SEP solely labels AMPA receptors on the cell surface, these videos should primarily display exocytosis events, excluding endosomal trafficking. Quantifying the number or frequency of these exocytosis events would provide a more direct assessment of AMPA receptor exocytosis. The "FRAP" recovery encompasses exocytosis, endocytosis, and lateral diffusion, which might not accurately capture alterations in exocytosis alone.

This is a very good point. Ideally, SEP tagged reagents are validated in cell culture using buffers of various pH levels. However, this is not possible in vivo in intact *C. elegans.* SEP::GLR-1 more than likely does not exclusively label surface receptors due to the rate of acidification of endosomes and the delay for pH quenching of SEP fluorescence (1-3 minutes; PMID: 29899033). Although infrequent, we do see SEP labeling of vesicles undergoing transport and perhaps very low ER signal. Based on short image streams (60 s) of SEP::GLR-1 following photobleaching, we are confident that the majority of our SEP::GLR-1 signal is at the synaptic surface. Specifically, we see only occasional transport (1-2 events per minute) and some faint exocytosis events that resulted in stable SEP::GLR-1 fluorescence (data not shown).

Although we agree that quantifying exocytosis events in videos would provide a better assessment of exocytosis only, this requires high imaging rates (20-50 fps) that result in a signal to noise ratio that is too low to reproducibly and consistently observe dim exocytosis events in vivo using SEP::GLR-1. Second, it is technically very difficult to locate the ideal focal plane where exocytosis happens at the membrane in vivo. SEP::GLR-1 FRAP is not a measure of only exocytosis but reflects the balance between exo- and endocytosis and is currently the only in vivo measurement of synaptic GLR-1 recruitment possible in our hands. We agree that this should more clearly written in the text and have now modified the manuscript to reflect this excellent observation from the reviewer. With the on-going development of new reagents and optical hardware, we plan to follow up with more precise experiments to assess if/how mitoROS individually affects exocytosis, diffusion, and endocytosis of GLR-1.

The authors observed an increase in raw fluorescence recovery for GLR-SEP in the MCU-1 mutant compared to the wild-type (Figure 2C), yet found that the percentage of recovery remained consistent in the mutant condition (Figure S2C). They inferred an elevation in exocytosis in the MCU-1 mutant based on these findings. However, there are concerns with this conclusion. Firstly, the FRAP experiments do not offer a direct assessment of exocytosis. Secondly, if the total expression level of GLR-SEP is augmented in the MCU-1 mutant, the raw fluorescence recovery could inherently be faster than that of the wild-type. The inclusion of appropriate controls is essential to validate these outcomes more robustly.

We do agree that the increase in basal GLR-1 in the *mcu-1* mutants makes interpreting the results of our FRAP experiments for *mcu-1(lf)* difficult. It is important to point out that basal GLR-1::GFP was not increased in *mcu-1(lf)* (data not shown) which suggests that the number of receptors at synaptic sites (either positioned in endosomal pools or at the membrane) is comparable between *mcu-1(lf)* and controls. When this result is considered alongside the increase in SEP::GLR-1 in *mcu-1(lf)*, it indicates that relatively more GLR-1 receptors are positioned at the membrane in *mcu-1(lf)*. We also agree with the reviewer that FRAP of GLR-1 encompasses more than exocytosis as mentioned above. We have also modified the text of the manuscript to use “synaptic recruitment” instead of exocytosis as mentioned at the end of our response to reviewer #3.

Figure 5B shows an enhanced % FRAP recovery for GLR-SEP in the control+PA Ru360 compared to the untreated group. However, Figure S2C doesn't indicate any difference between the Ru360 and untreated groups. Could the authors elucidate the inconsistency observed between these datasets?

For the SEP FRAP experiment for Figure 2, the background *C. elegans* strain (FJH 314, Appendix 1, Key Resources Table after figure legends) contained an integrated SEP::GLR-1 array allowing for more consistent expression with less inter-individual variability (which is ideal for quantifying steady-state GLR-1 levels such as in revised Figure 2-supplement 1B). Alternatively, for the SEP FRAP experiments in Figure 5 and 6 involving mitoKR, strain FJH 314 could not be used due to the co-expression of ChR2::mCherry which prevented us from validating the expression of mitoKR (which contains a red fluor) in this background. So, we had to conduct all SEP::GLR-1 FRAP experiments involving mitoKR on a different strain in which SEP::GLR-1 was expressed via an extrachromosomal array (FJH 635 and FJH 582 [containing mitoKR]). The strains used in Figure 2-supplement 1B had almost 3x higher SEP::GLR-1 compared to those used in Figure 5 (data not shown). Thus, calculating %FRAP by doing a 0-minute background subtraction followed by normalization to higher basal SEP::GLR-1 levels in the strain in Figure 2-supplement1B may have occluded the small effect of Ru360 treatment on %FRAP of SEP::GLR-1. We agree that this discrepancy should be better explained in the text, so we have mentioned that different SEP::GLR-1 expressing strains had to be used and that this may have led to inter-experimental inconsistencies. For additional transparency, we have also added the *C. elegans* strains used for each experiment to the main text and to all our figure legends.

There are discrepancies observed in the data presented. In Figure 2E and 2G, both FRAP recovery and trafficking events of GLR-GFP appear reduced in the MCU-1 mutant. Conversely, Figure 4E and 4H show a reduction in FRAP recovery and trafficking in the mitoKR condition. Given that ROS is diminished in the MCU-1 mutant but elevated during mitoKR global activation, can the authors elucidate and discuss these findings in greater detail? The current working model lacks clarity.

We thank the reviewer for pointing out the need for greater clarity in explaining our results for FRAP using GLR-1 GFP. Based on our mito-roGFP data, basal ROS levels at mitochondria in *mcu-1(lf)* or with Ru360 treatment are comparable to controls (see 0-minute stimulation groups in Figure 3E and 3G). Loss or inhibition of MCU-1 prevents the increase in mitoROS following neuronal stimulation which suggests that activity-dependent upregulation of mitoROS production is impacted by *mcu-1(lf)*/Ru360 but not basal mitoROS levels. So, based on this data, we cannot conclude that “ROS is diminished” in *mcu-1(lf).* Our data in revised Figure 6F suggest that the impact of *mcu-1(lf)* and mitoKR on GLR-1 transport involves parallel mechanisms since combining mitoKR and Ru360 doesn’t phenocopy Ru360 or mitoKR alone. We have modified our discussion of this and to our proposed model (illustrated in revised Figure 7) to more explicitly describe how our results suggest a paradoxical mechanism in which mitoROS and MCU-1 act within the same signaling pathway to regulate GLR-1 localization to the synaptic membrane (revised Figure 7B), but act in parallel signaling pathways in the neuronal soma to regulate GLR-1 export and dendritic transport (revised Figure 7A).

A clearer presentation might involve splitting the working model in Figure 6 into two distinct parts. One should depict the effect of ROS on AMPA receptor dendritic trafficking, while the other illustrates its influence on AMPA receptor surface expression/excocytosis. Given the potential differential impacts of mitochondrial calcium and ROS on these mechanisms, delineating the models for each process separately would enhance comprehension of the findings.

This is a great suggestion, and we agree that the illustration of our proposed model lacks mechanistic context. We have added an illustration to our model (revised figure 7A) to illustrate that our data indicates that in the cell body, MCU-1 and mitoROS act via separate signaling pathways to regulate GLR-1 transport .